**communications** engineering

# Towards inclusive risk-informed infrastructure development in expanding cities
Fabrizio Nocera[1,4], Yahya Gamal[2,3,4], Chenbo Wang[1] & Gemma Cremen [1] ✉

Conventional natural-hazard risk-modeling approaches do not consider possible unintended negative socioeconomic consequences of designing infrastructure expansions in a risk-informed way. Here, we propose a people-centered decision-making framework for urban infrastructure development that addresses this issue. The framework integrates a bespoke agent-based model that accounts for implications of variations in infrastructure expansion on dynamic land values and related residential location decision making. This means that the model captures macro-scale socioeconomic effects resulting from infrastructure development that are not explicitly related to natural-hazard events. The underlying algorithm balances these considerations with the successful operation of the infrastructure itself and the potential infrastructure performance losses that accompany a natural-hazard event. We demonstrate the framework by optimizing the expansion of transportation in a virtual urban testbed that imitates a typical expanding urban context in the Global South. This work can be used to guide inclusive risk-sensitive infrastructure planning in hazardous, rapidly growing cities.

The economic and social well-being of communities depend on the successful functionality of large-scale (typically intricately interdependent) critical infrastructure systems, including transportation, water and power supplies. However, the quality of critical infrastructure performance can be hampered by the occurrence of natural-hazard events (including those related to climate change), which can lead to notable societal impacts, including casualties, food insecurity, population displacement, business interruption, and unemployment[1,2]. These impacts are often more severe for individuals and communities in vulnerable and at-risk groups, e.g., those with low income and/or with disabilities[3], who typically have limited ability to cope with disaster-induced consequences[4]. Thus, it is crucial for relevant stakeholders to identify, assess, communicate, and appropriately manage potential natural-hazard-induced degradation in critical infrastructure performance[1,5].

A number of previous studies have developed engineering tools to quantify the impact of natural hazards on critical infrastructure in urban environments. Some have focused on simulating direct damage consequences to individual infrastructure components, like bridges, electric substations, and water and gas pipelines[6–8]. Other work has further translated component damage into changes in the functionality (i.e., indirect performance losses) of infrastructure systems, including transportation, power, water, and gas supplies[9–14]. For instance, studies of natural-hazard-induced functionality losses to transportation systems have centered on travel delay, unmet demand, and loss of connectivity[15–19], which have been useful for informing their risk-sensitive design[1,5,20,21]. Further work on these systems has focused on transforming functionality losses into broader societal consequences of hazard-induced disruption, including accessibility impacts to essential services/locations like hospitals, schools, and shelters[22–24] as well as effects on wellbeing experienced by people across different socioeconomic groups[25,26].

While these tools have helped to advance critical-infrastructure risk-modeling and design efforts, they have some limitations. Firstly, they typically overlook the context-specific bottom-up infrastructure performance needs of diverse users (including those from vulnerable communities) and focus instead on the top-down perspectives of infrastructure owners and operators[27,28]. This is a crucial shortcoming, given that people-centered infrastructure risk management approaches are becoming progressively more important in the context of climate change, rapid population growth, and increasingly interconnected urbanization[29,30]. Furthermore, current infrastructure risk-modeling approaches/models do not capture any unintended, undesirable socioeconomic consequences of optimizing infrastructure performance for hazard-induced impacts, such as gentrification or population segregation[31] that are not uncommon. For instance, improved connectivity that accompanied the development of China's expansive high-

[1]Department of Civil, Environmental and Geomatic Engineering, University College London, London, UK. [2]Urban Big Data Centre, University of Glasgow, Glasgow, UK. [3]Department of Geography, King's College London, London, UK. [4]These authors contributed equally: Fabrizio Nocera, Yahya Gamal. ✉e-mail: g.cremen@ucl.ac.uk

speed rail network resulted in rapid urbanization that unintentionally led to the loss of agricultural land[32] through haphazard urban sprawl[33]. Investments in bus and train systems within South Africa's Gauteng province, which aimed to create efficient and accessible urban transit, have reinforced spatial segregation[34]. These examples underscore the critical need to address and/or mitigate unintended socioeconomic consequences in infrastructure planning and expansion.

Here, we propose a people-centered, risk-informed decision-making framework for urban infrastructure development in growing cities that addresses current limitations in infrastructure risk modeling. The framework integrates an optimization procedure for expanding infrastructure that (i) balances its performance (in terms of bespoke user needs) in usual day-to-day conditions (i.e., before the occurrence of a natural-hazard event), as well as in the immediate aftermath of, and during the long-term period of recovery from, a (future) hazard event; while (ii) limiting (to an acceptable end-user-identified level) additional unintended gentrification-related socioeconomic consequences that may result from a purely performance-oriented infrastructure expansion. Consideration of (ii) is facilitated through the inclusion of an agent-based model (ABM) for residential location decision making[35–37] that quantifies both the benefit gained from and the affordability of a residential unit, given its proximity to the developed infrastructure[38–40].

We illustrate the proposed theoretical framework by designing an expansion of the road infrastructure serving the current population of Tomorrowville, a hypothetical community representing a Global South urban setting in terms of its socioeconomic and physical aspects (see the "Data description" section). The results of this work could be leveraged for holistic decision making on future infrastructure planning in hazardous, rapidly growing cities to ensure that infrastructure development is resilient to natural hazards yet not unintentionally non-inclusive.

## Methods

This section presents the proposed people-centered decision-making framework for future urban infrastructure expansion. The framework comprises two main calculation stages, as it couples infrastructure modeling (first stage) with a bespoke ABM (second stage) that accounts for implications of variations in infrastructure expansion on dynamic land values and related residential location decision making. The "Infrastructure modeling" section reviews infrastructure performance modeling that underpins the first stage (and the performance-oriented expansion as described in the "Results" section). The "Agent-based model for residential location decision making" section describes the ABM that represents the second stage. The "Holistic infrastructure development" section presents the end-to-end framework that integrates the two calculation stages and facilitates the holistic expansion described in the "Results" section.

### Infrastructure modeling

This section briefly reviews the mathematical formulation for modeling the time-varying performance of infrastructure using graph theory[12,14]. Graphs are mathematical structures representing the pairwise relations between objects called nodes (points or vertices) via edges (arcs, lines, or links.) We define a graph as $G = (V, E)$, where $V$ is the set of nodes and $E$ is the set of edges. Networks are graphs in which the nodes and edges also possess additional attributes like names, types, and state variables[14].

Infrastructure is represented as a collection of networks, where each network captures a specific feature or function of the infrastructure[14]. The collection of all networks is written as $\mathcal{G} = \{G^{[k]} = (V^{[k]}, E^{[k]}) : k = 1, \ldots, K\}$, where superscript $[k]$ is the feature or function captured by $G^{[k]}$. The state of each network is quantified at any given time using a unique set of vectors $[\mathbf{C}^{[k]}(t), \mathbf{D}^{[k]}(t), \mathbf{S}^{[k]}(t)]$ that represent the basic performance measures for $G^{[k]}$ - i.e., (i) capacity measures $\mathbf{C}^{[k]}(t)$, (ii) demand measures $\mathbf{D}^{[k]}(t)$, and (iii) supply measures $\mathbf{S}^{[k]}(t)$ - and are used to compute an overall performance measure $\mathbf{Q}^{[k]}(t)$ of $G^{[k]}$. In general, $\mathbf{C}^{[k]}(t)$, $\mathbf{D}^{[k]}(t)$, and $\mathbf{S}^{[k]}(t)$ depend on a set of variables $\mathbf{x}^{[k]}(t)$ that describe the

dynamic state of infrastructure accounting for deterioration[41] or repair[42], for instance.

We derive an overall infrastructure performance measure $Q(t)$ as an aggregate of the underlying network performances $\mathbf{Q}^{[k]}(t)$ that can be determined in various ways depending on the infrastructure of interest. For example, $Q(t)$ could be estimated from a topology-based approach in the case of road infrastructure[43] or from a flow-based approach in the case of potable water infrastructure[44].

$\mathfrak{R}[Q(t)]$ denotes some specific societal implication of the infrastructure performance, e.g., some measurement of distance between households to locations of interest related to a specific road infrastructure. In addition, $\mathfrak{R}[Q(t)]$ can be disaggregated based on socioeconomic factors (e.g., income, age, gender) to capture higher resolution effects of infrastructure performance (or non-performance) across different population segments.

### Agent-based model for residential location decision making

This section describes the ABM for residential location decision making[45,46], which is structured following Gamal et al.[47]. The ABM includes agents (buyers and sellers) that interact in a spatial context, where each agent represents one household. Here, buyers correspond to renters of residential units and sellers correspond to owners of residential units.

Following Alonso[36], the benefit an agent gains from (or an agent's attractiveness to) a residential unit is quantified using a utility-based approach, where utility is a function of residential unit attributes (e.g., distance from a specific location, access to water supply or sanitation infrastructure) and the individual agent's unique preferences towards such attributes. Utility is expressed as

$$U_{r,i} = \sum_{j=1}^{n} \alpha_{i,j} \cdot u(\lambda_{r,j}) \tag{1}$$

Where $U_{r,i}$ is the total utility of residential unit $r$ for the $i^{th}$ agent, $\lambda_{r,j}$ is a measurement of the $j^{th}$ attribute, $u(\lambda_{r,j})$ is an objective representation of the benefit associated with the $j^{th}$ attribute, $\alpha_{i,j}$ is the weight representing the subjective preference of the $i^{th}$ agent towards attribute $j$, and $n$ is the total number of attributes considered. $u(\lambda_{r,j})$ is written as

$$u(\lambda_{r,j}) = \begin{cases} \frac{\lambda_{r,j}}{\max_r(\lambda_{r,j})} & \text{if } \lambda_{r,j} \in \Lambda \\ 1 - \frac{\lambda_{r,j}}{\max_r(\lambda_{r,j})} & \text{otherwise} \end{cases} \tag{2}$$

where $\Lambda$ is the set of desirable attributes.

Agents are distinguished between buyers, $b$ and sellers, $s$, i.e., $i \in \{b, s\}$. The maximum price $b$ would pay for a residential unit is based on $U_{r,i}$; units with higher $U_{r,i}$ values (therefore higher $\lambda_{r,j}$ and/or $\alpha_{i,j}$) will yield a higher willingness to pay. Accordingly, the $b^{th}$ buyer's willingness to pay for the $r^{th}$ residential unit, $\text{WTP}_{r,b}$, is written as

$$\text{WTP}_{r,b} = \frac{H_{r,b} \cdot U_{r,b}^2}{\beta_{r,b} + U_{r,b}^2} \tag{3}$$

where $H_{r,b}$ is the available budget of the $b^{th}$ buyer, which can be expressed as a raw monetary value or a relative purchase capacity[48], and $\beta_{r,b}$ is a parameter controlling the convexity of $\text{WTP}_{r,b}$ that reflects the risk appetite of the buyer. The range of $\beta_{r,b}$ is the same as that of $U_{r,b}$, where high values of $\beta_{r,b}$ indicate risk-averse behavior and low values indicate risk-taking behavior.

The rental price of the $r^{th}$ residential unit set by the $s^{th}$ seller, $P_{r,s}$ is based on the benefit of the unit to the seller $U_{r,s}$, and is expressed as

$$P_{r,s} = \frac{H_{r,s} \cdot U_{r,s}^2}{\beta_{r,s} + U_{r,s}^2} \tag{4}$$

where $H_{r,s}$ is the buyer's budget as perceived by the seller, analogous to $H_{r,b}$, and $\beta_{r,s}$ is analogous to $\beta_{r,b}$.

**Modeling details.** The ABM represents agents' behaviors in the form of relocations, which occur when a buyer can no longer afford to pay their rent (i.e., $P_{r,s} > \mathrm{WTP}_{r,b}$). Relocations are triggered by changes in $P_{r,s}$ and/ or $\mathrm{WTP}_{r,b}$, which are the result of changes in $\lambda_{r,j}$ that ultimately stem from the infrastructure development process. The number of triggered relocations $\varepsilon$ due to changes in (expansions of) the infrastructure (i.e., the number of times $P_{r,s} > \mathrm{WTP}_{r,b}$) can be considered a proxy for gentrification and represents the unintended socioeconomic consequences of infrastructure development in this study.

Residential location decision making is modeled by assigning a choice of $\theta_N^*$ vacant residential units to relocating renters in their current neighborhood $N$. Relocating renters move to the closest (vacant) residential unit $r^*$ within $\theta_N^*$ that satisfies the following conditions: (i) $\mathrm{WTP}_{r^*,b} \geq P_{r^*,s}$ and (ii) $U_{r^*,b} \geq U_b^*$, where $U_b^*$ is the average utility of the $\theta_N^*$ residential units. If none of the $\theta_N^*$ residential units satisfy these conditions, an alternative neighborhood within the urban area is randomly selected and the same process of identifying a satisfactory residential unit (or another alternative neighborhood) is repeated. If no satisfactory residential unit is found, the relocating renters emigrate from the urban system and the affected household is no longer considered in the analysis.

## Holistic infrastructure development

This section presents the proposed end-to-end approach for achieving a holistic infrastructure expansion that is both performance-oriented (risk-informed) and accounts for unintended socioeconomic consequences of infrastructure development. First, we leverage the theory of the "Infrastructure modeling" section to formulate the performance-oriented expansion as an optimization problem, where the objective function accounts for infrastructure performance during day-to-day operations, in the immediate aftermath of a disrupting event (i.e., response phase), and during the long-term recovery phase. Then, we discuss the approach for solving the optimization problem while accounting for unintended consequences that are quantified using the utility-based residential location ABM in the "Agent-based model for residential location decision making" section.

**Mathematical formulation.** The objective function is expressed as

$$Z = \max \mathbb{E}\left[\left(\gamma_1 \cdot Z_1 + \gamma_2 \cdot Z_2 + \gamma_3 \cdot Z_3\right)\right] \tag{5}$$

where $\mathbb{E}[\cdot]$ is the expected value operator, $\gamma_1$, $\gamma_2$, and $\gamma_3$ are weights that respectively control the relative importance of infrastructure performance on a day-to-day basis ($Z_1$), during the immediate post-hazard response period ($Z_2$), and the longer term recovery phase ($Z_3$). $\gamma_1$, $\gamma_2$, and $\gamma_3$ values are defined in consultation with relevant stakeholders, in a participatory, people-centered approach to risk-informed decision making. $Z_1$ is written as

$$Z_1 = \frac{1}{n_a} \sum_{a=1}^{n_a} \omega_a \frac{1}{N_{\mathrm{H}}} \sum_{i=1}^{N_{\mathrm{H}}} w_i \mathfrak{R}_{i,a}\left[Q(t_{0^-}, \mathbf{g})\right] \tag{6}$$

where $n_a$ is the number of considered infrastructure needs (types), $\omega_a$ is the weight (priority) placed on the $a^{th}$ infrastructure need, $N_{\mathrm{H}}$ is the number of household agents in the community, and $w_i$ is the weight (priority) placed on meeting the $i^{th}$ household's infrastructure needs. $\mathfrak{R}_{i,a}\left[Q(t_{0^-}, \mathbf{g})\right]$ describes a specific implication of infrastructure performance at household-level during $t_{0^-}$ (before the occurrence of the hazard event) and $\mathbf{g}$ is the set of $G^{[k]}$ to be added as part of the infrastructure expansion. For example, in the case of a topology-based analysis of road infrastructure that is used for accessing hospitals, schools, and workplaces, $\mathfrak{R}_{i,a}\left[Q(t_{0^-}, \mathbf{g})\right]$ for the $i$th

household is written as

$$\mathfrak{R}_{i,a}\left[Q(t_{0^-}, \mathbf{g})\right] = \frac{\eta_{i,a}^{(H)}(t_{0^-}, \mathbf{g})}{\eta_{i,a}^*(t_{0^-}, \mathbf{g})} = \frac{1}{N(i)} \sum_{m=1}^{N(i)} \frac{d_{i,a(m)}^*}{d_{i,a(m)}} \tag{7}$$

where $N(i)$ is the number of individuals in household $i$ that have infrastructure need $a$ (i.e., access to a hospital, school, or workplace) and $d_{i,a(m)}$ is the distance from the residence of household agent $i$ to the activity (location) of interest of the $m^{th}$ individual in household $i$. $\eta_{i,a}^*(t_{0^-}, \mathbf{g})$ is a reference value for normalizing $\eta_{i,a}^{(H)}(t_{0^-}, \mathbf{g})$ such that $d_{i,a(m)}^*$ is a corresponding value in terms of road distance, enabling each component of the objective function in Eq. (5) to be added together. In the limiting case when the distance between origin and destination is infinity (i.e., the destination is unreachable), $\eta_{i,a}^{(H)}(t_{0^-}, \mathbf{g}) = 0$.

$Z_2$ is expressed as

$$Z_2 = \frac{1}{n_{a'}} \sum_{a'=1}^{n_{a'}} \omega_{a'} \frac{1}{N_{\mathrm{H}}} \sum_{\substack{i=1, \\ p(i) \subseteq i \in \Omega_{a'}}}^{N_{\mathrm{H}}} w_i \mathfrak{R}_{i,a'}\left[Q(t_{0^+}, \mathbf{g})\right] \tag{8}$$

where $\omega_{a'}$ is the weight of the $a'^{th}$ infrastructure need in the response phase $t_{0^+}$, $p(i) \subseteq i \in \Omega_{a'}$ identifies the individuals (in household $i$) associated with the $a'^{th}$ infrastructure need, and $\mathfrak{R}_{i,a'}\left[Q(t_{0^+}, \mathbf{g})\right]$ describes some aspect of infrastructure performance in $t_{0^+}$. For example in the case of road infrastructure, $a'$ may refer to accessing hospitals (for immediate treatment) or shelters (if there is post-event dislocation) and $p(i) \subseteq i \in \Omega_{a'}$ would define the individuals in the $i^{th}$ household that are either injured or displaced. In the context of using a topology-based approach to measure the performance of road infrastructure, $\mathfrak{R}_{i,a'}\left[Q(t_{0^+}, \mathbf{g})\right]$ is defined as

$$\mathfrak{R}_{i,a'}\left[Q(t_{0^+}, \mathbf{g})\right] = \frac{\eta_{i,a'}^{(H)}(t_{0^+}, \mathbf{g})}{\eta_{i,a'}^{(H)}(t_{0^-}, \mathbf{g})} \tag{9}$$

capturing the household's increase in distance to each location of interest at $t_{0^+}$ compared to $t_{0^-}$.

$Z_3$ is expressed as

$$Z_3 = \frac{1}{T_{\mathrm{R}}} \sum_{\tau=t_{0^+}}^{T_{\mathrm{R}}} \frac{1}{n_a} \sum_{a=1}^{n_a} \omega_a \frac{1}{N_{\mathrm{H}}} \sum_{i=1}^{N_{\mathrm{H}}} w_i \mathfrak{R}_{i,a}\left[Q(\tau, \mathbf{g})\right] \tag{10}$$

where $T_{\mathrm{R}}$ represents the time at which recovery activities are completed, and $\mathfrak{R}_{i,a}\left[Q(\tau, \mathbf{g})\right]$ is a dynamic measure of some aspect of infrastructure performance related to the $i$th household during the recovery process. In the case of a topology-based analysis of road infrastructure, $\mathfrak{R}_{i,a}\left[Q(\tau, \mathbf{g})\right]$ is expressed as

$$\mathfrak{R}_{i,a}\left[Q(t, \mathbf{g})\right] = \frac{\eta_{i,a}^{(H)}(t, \mathbf{g})}{\eta_{i,a}^{(H)}(t_{0^-}, \mathbf{g})} \tag{11}$$

and is analogous to $\mathfrak{R}_{i,a'}\left[Q(t_{0^+}, \mathbf{g})\right]$, where the locations of interest are the same as those captured by $\mathfrak{R}_{i,a}\left[Q(t_{0^-}, \mathbf{g})\right]$ in Eq. (7). Note that while $\mathfrak{R}_{i,a^*}[.]$ have been described in terms of a topology-based approach for quantifying road infrastructure performance, the proposed formulation is general enough for application to any infrastructure and performance measurement approach[44,49].

The constraints of the optimization are now presented. The first constraint is

$$C_{\mathrm{P}} \leq M_{\mathrm{P}} \tag{12}$$

where $C_p$ is the cost of implementing a specific infrastructure expansion and $M_p$ is the budget allocated to the infrastructure development process. The resulting $G^{[k]}$ must be a connected network, i.e.,

$$\forall v_1, v_2 \in V^{[k]}, \exists \varphi(v_1, v_2) \tag{13}$$

where $v_1$ and $v_2$ are two generic nodes in $V^{[k]}$, and $\varphi(v_1, v_2)$ is a path between them. The degree of nodes in the resulting $G^{[k]}$ must be less than a specified threshold to avoid impracticable infrastructure expansion (e.g., an intersection created by more than four roads in the case of road infrastructure), written as

$$\Delta\left(G^{[k]}\right) \leq \delta\left(G^{[k]}\right) \tag{14}$$

where $\Delta\left(G^{[k]}\right)$ is the maximum degree of $G^{[k]}$ and $\delta\left(G^{[k]}\right)$ is a corresponding specified threshold. Additional non-negative constraints are

$$\omega_a \geq 0, \forall a \tag{15}$$

$$w_i \geq 0, \forall i \tag{16}$$

$$\omega_{a'} \geq 0, \forall a' \tag{17}$$

$$\gamma_1, \gamma_2, \gamma_3 \geq 0 \tag{18}$$

Weights $\omega_a$, $w_i$, and $\omega_{a'}$, and $\gamma_1$, $\gamma_2$ and $\gamma_3$ must sum to one, written as

$$\sum_{a=1}^{n_a} \omega_a = 1 \tag{19}$$

$$\sum_{i=1}^{N_H} w_i = 1 \tag{20}$$

$$\sum_{a'=1}^{n_{a'}} \omega_{a'} = 1 \tag{21}$$

$$\gamma_1 + \gamma_2 + \gamma_3 = 1 \tag{22}$$

The final constraint of the optimization is

$$\frac{\eta_{k,a}^{(H)}(\tau^*, \mathbf{g})}{\eta_{k,a}^{(H)}(t_{0^-}, \mathbf{g})} \geq \xi(\tau^*), \forall \tau^* \tag{23}$$

where $\xi(\tau^*)$ represents a lower threshold for infrastructure performance at time $\tau^*$, facilitating a requirement for the infrastructure to be restored to pre-hazard performance levels within a certain period from the occurrence of the hazard event.

**Obtaining the final holistic expansion.** The formulation described in the "Mathematical formulation" section can be classified as a combinatorial optimization problem, where the optimal infrastructure layout (topology) results from a finite set of possible infrastructure interventions, i.e., added edges, such as new roads. Figure 1 summarizes the workflow that is used to solve for the final holistic infrastructure expansion.

First, an augmented infrastructure layout is defined that includes the existing infrastructure layout and the full set of potential (candidate) edges for development. Several procedures can be used to obtain the augmented layout, such as (i) identifying bespoke candidate edges on a context-specific basis in consultation with relevant stakeholders and manually digitizing them in a geographic information system, (ii) using digitized geospatial data to define a regular grid of points and finding the least cost paths among them or (iii) using a fully-automated interactive procedural modeling approach based on tensor field theory[50]. Then, the subset of candidate edges to be

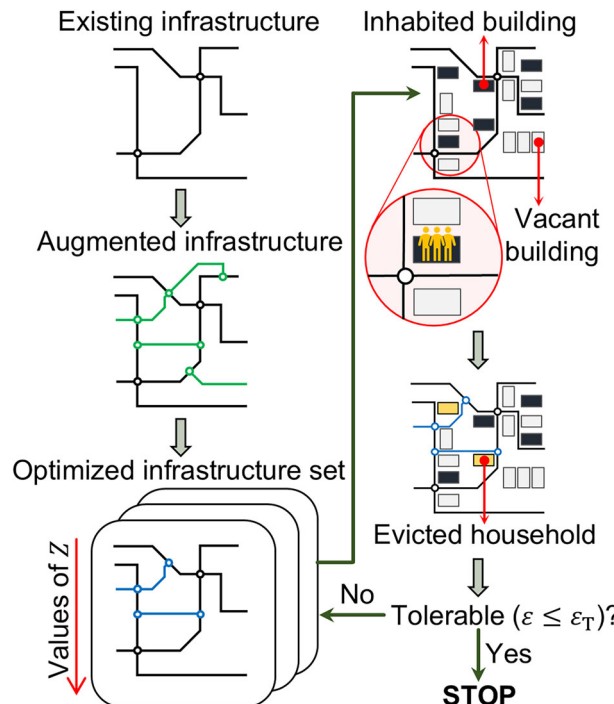

**Fig. 1 | Workflow for obtaining the final infrastructure expansion.** $Z$ is the objective function of the performance-oriented infrastructure expansion optimization problem. $\varepsilon$ is the number of relocations triggered by infrastructure expansion. $\varepsilon_T$ is an end-user-specific acceptable level of triggered relocations.

added to the existing infrastructure maximizes the value of $Z$ in Eq. (5) and satisfies the constraints outlined in the "Mathematical formulation" section.

However, there are computational complexities associated with this optimization problem, including nonlinearity, nonconvexity, and non-differentiability of $Z$ in Eq. (5). Consequently, an exhaustive search is not feasible, and a heuristic approach must be used to obtain a near-optimal solution instead. Our proposed heuristic approach involves a simulated annealing-based metaheuristic procedure. Other metaheuristics, like genetic algorithms, ant colony systems, and tabu searches, could be used instead, but they typically perform worse than simulated annealing for this type of optimization problem[51–53].

The search begins by randomly selecting a subset of candidate edges that satisfy the constraints of the "Mathematical formulation" section and using the corresponding value of $Z$ as the initial optimization solution. The simulated annealing-based metaheuristic procedure then maximizes $Z$ by applying small (random) changes to the decision variable $\mathbf{g}$, i.e., the edges to be added from the full candidate list. Each small perturbation involves randomly selecting one of the following actions: *add*, *remove*, or *replace*. If *add* is selected, a new edge candidate is randomly added to the current subset from the full candidate list. If *remove* is selected, a random candidate is removed from the current subset. If *replace* is selected, a candidate from the current subset is randomly substituted with a new candidate selected from the full set of candidates. If a neighboring solution (resulting from the perturbation) improves the value of $Z$, a further search starts in the neighborhood of this point. If an improved solution cannot be found, the current solution is accepted with a certain probability, i.e., $\exp(-Z/T)$ where $T$ is one of the hyperparameters of the optimization search algorithm, typically known as the temperature. Infeasible solutions that violate the constraints of the "Mathematical formulation" section are avoided by adding a dynamic penalty function to the solution of the objective function[54], such that Eq. (5) is rewritten as

$$Z' = \begin{cases} Z & \text{if } \mathbf{g} \in \Omega_f \\ Z + P(\mathbf{g}) & \text{otherwise} \end{cases} \tag{24}$$

where $\Omega_f$ represents the set of feasible solutions and the penalty function $P(\mathbf{g})$ is introduced when any constraint of the "Mathematical formulation" section is violated.

At the end of the search, the optimization results are compiled in a list of infrastructure development layouts (expansions) ranked in terms of $Z$ (equivalent to the optimized infrastructure set shown in Fig. 1). The performance-oriented expansion is then the highest-ranked layout of the set. Changes in the infrastructure expansion lead to variations in $\lambda_{r,j}$ and therefore $u(\lambda_{r,j})$ in Eq. (2), which ultimately produce changing values of $\varepsilon$. The final holistic infrastructure expansion is the one with the highest value of $Z$ that also satisfies $\varepsilon \leq \varepsilon_T$, where $\varepsilon_T$ is a pre-determined, end-user-specific acceptable level of unintended consequences (i.e., triggered relocations).

## Data description

**Case study.** We use the proposed framework for designing an expansion of the road infrastructure in the 500-ha virtual urban testbed of Tomorrowville. Tomorrowville was designed to represent a typical Global South urban setting based on Nairobi (Kenya) and Kathmandu (Nepal) data[55]. The testbed is a geospatial database of urban features that includes information on land use, building and infrastructure (physical) characteristics, household (social) characteristics (such as income levels), and individual (social) characteristics, as well as detailed data on each person's daily infrastructure needs, which include access (proximity) to hospitals, schools, and workplaces. We use TV0 in this study, which represents the current urban layout of Tomorrowville (more details can be found in Menteşe et al.[55]). TV0 contains a total of 4810 buildings and 7809 households (all assumed to be renters), of which 4236 are low-income, 1705 are mid-income, and 1868 are high-income. The network representing the existing road infrastructure of TV0 contains 1128 edges and 999 nodes.

**Data for performance-oriented infrastructure expansion.** The hypothetical stakeholders are assumed to (i) consider infrastructure need in terms of accessibility (proximity) to hospitals ($h$), schools ($e$), and workplaces ($l$), (ii) regard each infrastructure need as being equally important regardless of time, (iii) place equal importance on day-to-day and immediate post-hazard infrastructure performance but to not consider the long-term recovery process when making infrastructure expansion decisions; and (iv) hold a pro-poor vision on future urban expansion, in line with the latest thinking on disaster risk management and assessment[56,57] as well as the guiding principles of the Sendai Framework for Disaster Risk Reduction[58]. These views are reflected in the following input parameter values used: (i) $\gamma_1 = \gamma_2 = 0.5$, (ii) $\gamma_3 = 0$, (iii) $\boldsymbol{\omega}_a = \boldsymbol{\omega}_{a'} = [1/3, 1/3, 1/3]$, and (iv) $\mathbf{w}_i = [0.7, 0.2, 0.1]$, where the vector entries respectively refer to low-income, mid-income, and high-income households. We assume a road is impassible if it is exposed to a flood water height of more than 0.3 m[59], which leads to decreased values of $\eta_{i,a'}^{(H)}(t_{0^+}, \mathbf{g})$ and therefore $\mathfrak{R}_{i,a'}[Q(t_{0^+}, \mathbf{g})]$ in Eq. (9). We further assume $C_p$ is equal to £5000/m, $M_p$ is £70M, and $\varepsilon_T = 750$. The augmented expansion of the road infrastructure is obtained through manual digitization of the candidate edges, hypothetically reflecting the outcome of a conversation with potential stakeholders. The resulting augmented network contains 1740 edges and 1483 nodes.

**ABM data for unintended consequence quantification.** We assume $\beta_{r,b} = \beta_{r,s} = 1$ for all agents, reflecting a risk-neutral outlook. $\lambda_{r,j}$ represent undesirable attributes, comprising the road distance from each household's residential unit to hospitals $\lambda_{r,1}$, schools $\lambda_{r,2}$, and workplaces $\lambda_{r,3}$ (only for households with working individuals, such that $n = 2$ or $u(\lambda_{r,3}) = 0$ otherwise), in line with the infrastructure needs previously identified. Note that these distances assume normal day-to-day infrastructure performance, such that a household's willingness to pay for a residential unit does not account for natural-hazard-induced travel disruptions. No desirable attributes are considered. We further assume $\alpha_{b,1} \sim \mathrm{Uniform}(0, 1)$, $\alpha_{b,2} \sim \mathrm{Uniform}(0, 1)$, $\alpha_{b,3} \sim \mathrm{Uniform}(0, 1)$, $\alpha_{s,1} \sim \mathrm{Uniform}(0, 1)$, $\alpha_{s,2} \sim \mathrm{Uniform}(0, 1)$ and $\alpha_{s,3} = 0$, i.e., there are generally heterogenous, independent preferences towards the various considered locations, except owners of residential units do not value the distance of their property (that they do not live in) to work. In the absence of more relevant data, $H_{r,i}$ values are based on information collected from Greater Cairo, Egypt[47], which is deemed acceptable in this case given that Tomorrowville is designed to represent a general Global South urban setting. These values are quantified in terms of relative purchasing capacity that is measured over a continuum scale based on (i) the relative proportion of wealth/income across households in low-, mid-, and high-income groups[47,60] and (ii) the absolute accumulated wealth/income associated with each income group[61,62]. This means that $H_{r,b} \sim \mathrm{Uniform}(a_b, b_b)$ - where $\{a_b, b_b\} = \{0, 0.9\}$ for low-income households, $\{a_b, b_b\} = \{0.9, 1.96\}$ for mid-income households, and $\{a_b, b_b\} = \{1.96, 4.1\}$ for high-income households - and $H_{r,s} \sim \mathrm{Uniform}(a_s, b_s)$ - where $\{a_s, b_s\} = \{0, 0.93\}$ for low-income households, $\{a_s, b_s\} = \{0.93, 1.8\}$ for mid-income households, and $\{a_s, b_s\} = \{1.8, 2.5\}$ for high-income households - such that $a_i$ and $b_i$ represent a scaled ratio between the minimum/maximum income of a given income and agent group and the maximum income of the richest corresponding group. We perfom Monte Carlo sampling of the probability distributions to produce 100 sets of each uncertain input variable per household and compute an expected value of $\varepsilon$, $E(\varepsilon)$, which is then used to determine the final holistic solution (replacing $\varepsilon$).

**Flood hazard model.** The hazard event considered is similar to the 25-year mean return period pluvial fluvial flooding event presented in Jenkins et al.[63]. The flood simulations are generated using CAESAR-Lisflood, a model that combines the Lisflood-FP hydrological and surface flow model[64] with the CAESAR landscape evolution model[65]. The discharge and rainfall time series are generated using moderate to peak daily data based on the Department of Hydrology and Meteorology, Nepal records, such that simulations are consistent with the Tomorrowville topography. More details on flood modeling for Tomorrowville can be found in Jenkins et al.[63].

## Results

The Tomorroville road infrastructure expansion is designed considering a 25-year mean return period pluvial fluvial flooding event and the urban development vision of a hypothetical set of Tomorrowville stakeholders. The design involves selecting an optimal finite set of roads from an augmented list of regularly laid out candidate edges, based on various constraints (see the "Methods" section and Fig. 2) that include a stakeholder-defined acceptable limit of $\varepsilon_T = 750$ Tomorrowville households (less than 10%) evicted due to rent price increases triggered by the developed infrastructure. Higher priority is placed on satisfying the road infrastructure requirements of low-income households, in line with the pro-poor vision for urban development held by the hypothetical stakeholders. We first present the road expansion that results if only the performance of the infrastructure is accounted for (i.e., unintended consequences related to $\varepsilon_T$ are ignored), which is herein described as the performance-oriented expansion. We then present the final holistic expansion that also complies with $\varepsilon_T$.

### Performance-oriented expansion

Figure 3 (left panel) displays the performance-oriented expansion of the Tomorrowville road infrastructure. In this case, the final infrastructure layout results in $Z = 0.288$ (where $Z$ is a performance measurement; see Eq. 5 in the "Methods" section), which translates into a post-flood loss of connectivity (caused by impassable roads creating a theoretical travel distance of infinity) to hospitals, schools, and workplaces for 1684, 1414, and 3722 households, respectively. These values respectively represent 78%, 20%, and 27% reductions compared to those obtained for the existing road infrastructure in Tomorrowville. Figure 4 provides the number of households with no access to hospitals, schools, and workplaces after the considered flood event, disaggregated by income level. The expanded road

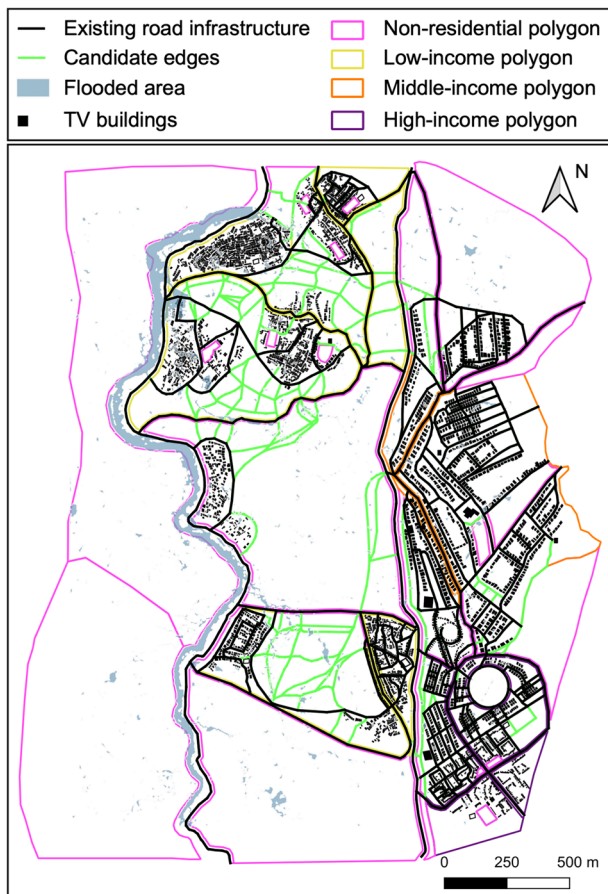

**Fig. 2 | Tomorrowville road infrastructure (existing and candidate edges for expansion) and considered flooding event.** Adapted from Menteşe et al.[55] published under a CC BY 4.0 licence (https://creativecommons.org/licenses/by/4.0/).

infrastructure leads to a greater (absolute) reduction in post-flood accessibility losses for low-income households compared to mid- and high-income groups, reflecting the pro-poor stakeholder vision. However, this infrastructure layout would lead to $E(\varepsilon) = 919$ evicted households (all low-income), which is more than 20% greater than $\varepsilon_{\mathrm{T}}$.

### Holistic expansion

Figure 3 (right panel) presents the holistic expansion of the Tomorrowville road infrastructure that ensures $E(\varepsilon) \leq \varepsilon_{\mathrm{T}}$. This infrastructure layout results in $Z = 0.277$ (see Eq. 5 in the "Methods" section), which translates into a post-flood loss of accessibility to hospitals, schools, and workplaces for 1769, 1499 and 4465 households, respectively. These values represent 77%, 15%, and 13% reductions compared to those obtained for the existing road infrastructure in Tomorrowville.

Figure 4 contrasts these results with those obtained for the performance-oriented expansion. The holistic infrastructure expansion reduces $E(\varepsilon)$ and $Z$ by 24.3% and 3.8%, respectively, compared to the performance-oriented expansion. In other words, the framework can produce an infrastructure expansion with tolerable unintended consequences at the expense of only slightly sub-optimal performance. The reduction in $Z$ translates into a minor rise (~5%) in post-flood accessibility losses for low-income households. A more substantial rise in workplace accessibility losses is obtained for mid- (14.5%) and high-income (47.8%) households. These observations specifically highlight the value of adopting a well-rounded perspective in a deliberately pro-poor development process, i.e., gaining a substantial reduction in the number of evicted low-income households at the expense of accepting noticeable lower infrastructure performance for those more well off.

Note that further investigation of the relationship between prices, infrastructure performance, and triggered relocations across different income groups can be found in the Supplementary Notes 1 and 2.

## Discussion

This paper proposed a people-centered, risk-informed decision-making framework for infrastructure development in growing cities. The framework extends beyond conventional natural-hazard infrastructure impact assessments by facilitating external participation in the design process and holistically accounting for unintended consequences of risk-informed infrastructure development (gentrification), recognizing that equitable development may ultimately require a departure from a strictly performance-driven outlook.

We formulated the infrastructure development process as a combinatorial optimization problem (see the "Methods" section), in which the objective is to maximize the performance of the infrastructure according to bespoke stakeholder (end-user) priorities and needs in three distinct temporal phases, i.e., business-as-usual conditions, in the immediate aftermath of a (future) hazard event, and during the long-term post-event recovery process. The final infrastructure expansion is one that also leads to an acceptable level of unintended socioeconomic (gentrification-related) consequences, which are quantified using a bespoke agent-based model that captures the implications of variations in infrastructure development on land values and resulting dynamic residential location decision making. The holistic, inherently participatory nature of the proposed framework can help to prioritize the needs of lower-income populations and generally support a pro-poor approach in future risk-informed urban development[56]. As such, the framework is cross-cutting, addressing broad sustainable development goals (i.e., Sustainable Development Goal 10: *Reduced inequalities*) as well those that are more specifically focused on engineered assets (i.e., Sustainable Development Goal 9: *Build resilient infrastructure, promote inclusive and sustainable industrialization and foster innovation*; Sustainable Development Goal 11: *Make cities and human settlements inclusive, safe, resilient and sustainable*).

The case study demonstrated that the framework could produce an infrastructure expansion with tolerable unintended consequences (specifically benefitting low-income households) at the expense of only slightly sub-optimal performance (predominantly impacting mid- and high-income households). Sensitivity analyses involving some of the case study parameters (see Supplementary Notes 1 and 2) confirmed a complicated relationship between infrastructure performance and gentrification; better infrastructure performance does not necessarily entail an additional gentrification cost, underlining the importance of explicitly tracking both variables for informed decision making. Furthermore, these analyses revealed that low-income households would remain disproportionately susceptible to triggered relocations (and therefore would still benefit the most from a gentrification cap) even if rental prices increased somewhat (without considering any infrastructure expansion). It is important to note that these conclusions are based on one example involving the road infrastructure of a Global South virtual testbed in the presence of flooding, where there is a strong spatial correlation between income and level of hazard exposure. If more higher income households resided in flood-prone areas, the extent and pattern of gentrification would differ in line with the corresponding offset in the (generally larger) prices and values of willingness to pay; we do not investigate this hypothetical scenario in detail, because the strong spatial relationship between income and hazard exposure that exists within Tomorrowville has been deliberately designed to mirror real-world contexts[57]. Further testing of the framework's capabilities in real-world settings is required; the proposed formulation is general enough for application to any hazardscape and critical infrastructure challenge of interest (e.g., from siting electric vehicle chargers to constructing seawalls) involving a trade-off between enhancing the performance of the infrastructure and limiting triggered relocations (i.e., gentrification) caused by resulting price increases.

**Fig. 3 | Different expansions of Tomorrowville road infrastructure.** Performance-oriented expansion (left panel). Holistic expansion (right panel). Adapted from Menteşe et al.[55] published under a CC BY 4.0 licence (https://creativecommons.org/licenses/by/4.0/).

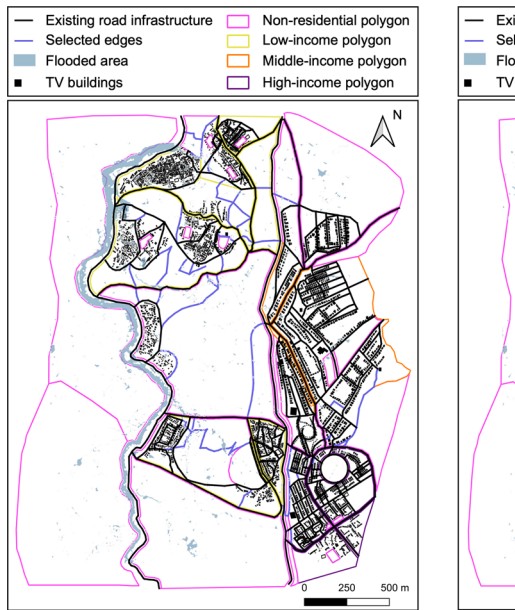
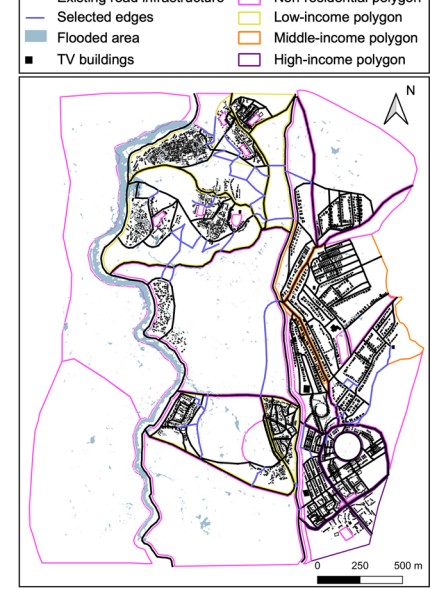

**Fig. 4 |** Number of households with no access to hospitals, schools, and workplaces for the existing road infrastructure in Tomorrowville (baseline), the performance-oriented expansion, and the holistic expansion, disaggregated by income level.

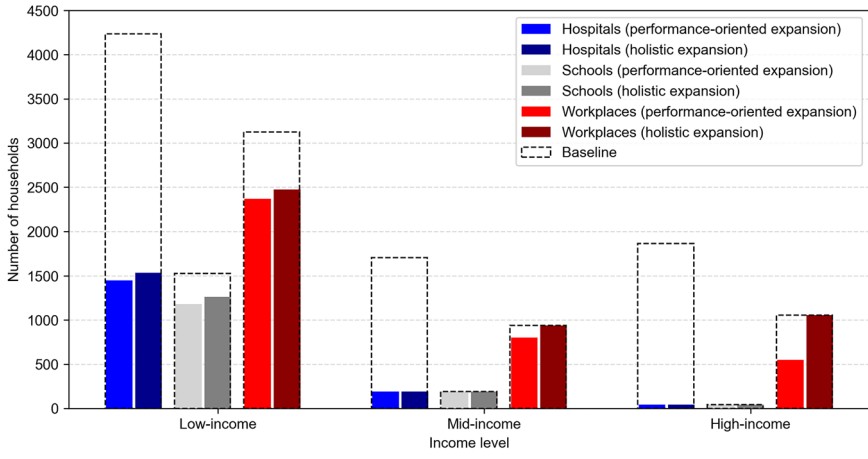

This study features some limitations and simplifying assumptions that warrant comment. First, infrastructure performance is quantified using a topology-based (connectivity-based) approach. While connectivity is generally a necessary condition for a fully operational infrastructure, flow-based approaches[24,49] - which track the transfer of specific quantities of interest across infrastructure in space and time—more accurately capture the ability of infrastructure to provide essential goods and services. However, the proposed framework is flexible enough to accommodate flow-based approaches in subsequent applications (through appropriate modification of Eqs. (7), (9), and (11) in the "Methods" section).

Second, the case study application relies on hypothetical stakeholder inputs, assuming that infrastructure needs are limited to accessing hospitals, schools, and workplaces, and that a household's willingness to pay is exclusively determined by the travel distance to these critical infrastructure under normal conditions (i.e., disruptions from hazards are not considered to influence the perceived value of a given residence). While it is realistic to assume that residences with greater connectivity to important locations are valued higher[66–68], such that infrastructure development in hazard-prone areas could force local low-income households out of their homes, it may not be reasonable to presume that the attractiveness of better connected areas to higher income households is independent of exposure to natural

hazards. However, tracking household movements beyond triggered relocations is outside the scope of this study.

Real-world stakeholder participation in the infrastructure development process may involve a series of local engagement workshops that leverage targeted surveys to determine infrastructure needs and their relevant importance, balancing different perspectives across diverse groupings. Potential participants include (future) residents within the urban setting of interest, planners, government representatives, engineering professionals, as well as experts and researchers in urban disaster risk. These types of workshops have been successfully used for similar real-life decision-making processes. For instance, Tompkins et al.[69] held workshops with various stakeholders (e.g., local council representatives, coastal management groups, coastal businesses, and residents) to inform long-term coastal planning for climate change in Christchurch Bay, England, and the Orkney Islands, Scotland. Similarly, Bostick et al.[70] leveraged the results of a stakeholder workshop (attended by members of city and county emergency management, Gulf Coast and Mobile Bay environmental management spokespeople, port industry representatives, and public works employees) to prioritize a set of risk management initiatives aimed at enhancing coastal resilience in Mobile Bay, Alabama (USA). Other people-centered data required (e.g., on residents' socioeconomic characteristics and preferences

in terms of infrastructure attributes) could be collected following the interview- and survey-based approach proposed in Gamal[47].

The case-study application also neglects the long-term post-event recovery phase. Precisely characterizing infrastructure recovery during this period would involve knowledge of context-specific disaster recovery priorities and infrastructure recovery scheduling details, such as typical construction activity precedence and workforce availability over large geographic areas[71]. Furthermore, the case study makes use of simplified economic assumptions regarding the cost of road construction and the available budget for infrastructure development, given its hypothetical nature; this information should be readily accessible for any real-life infrastructure expansion project, however.

In conclusion, the proposed framework represents a paradigm shift in risk-sensitive decision support on urban infrastructure development. It paves the way for the creation of resilient yet equitable future cities.

## Data availability
Data underlying the results presented in this paper may be obtained from the corresponding author upon request.

## Code availability
Codes underlying the results presented in this paper are publically available in the following repositories: https://github.com/YahyaGamal/Infrastructure_optimisation_ABM and https://github.com/YahyaGamal/Communications_paper_graphs.

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

## Acknowledgements

This work was supported in part by the UKRI GCRF under grant NE/S009000/1, Tomorrow's Cities Hub. The opinions and findings presented are those of the authors and do not necessarily reflect the views of the sponsor. The authors are also grateful to Dr. Carmine Galasso, Dr. Ryerson

Christie, Dr. Maria Evangelina Filippi, and Dr. Tom O' Shea for helpful feedback on parts of this study.

## Author contributions
F.N., Y.G., and G.C. conceived and designed the research. F.N. and G.C. developed the theoretical people-centered risk-informed framework. F.N. carried out the infrastructure performance analyses. Y.G. developed and implemented the agent-based model for residential location decision making. C.W. contributed to the literature review. G.C. provided project supervision. All authors contributed to writing and discussions on the paper.

## Competing interests
The authors declare no competing interests.

## Additional information

Primary Handling Editors: Mengying Su, Miranda Vinay, Rosamund Daw. [A peer review file is available].

