## [Transparent Peer Review file · Communications Engineering]

Towards inclusive risk-informed infrastructure development in expanding cities

Corresponding Author: Dr Gemma Cremen

Version 0:

Reviewer comments:

Reviewer #1

(Remarks to the Author)

The manuscript is interesting, timely, and overall well-written. I recommend publication after the following comments are addressed:

1. The stated limitations of the current approaches (Lines 46-49) are somewhat unfairly representing the strengths of the literature and existing approaches. It's not clear why the existing approaches would not be suitable for future hazards, and the statement that they ignore the context/stakeholder priorities is at least too strong. See for example <https://doi.org/10.1080/23789689.2018.1448667>, <https://doi.org/10.1111/j.1539-6924.2006.00801.x>, and <https://doi.org/10.1061/AJRUA6.00009> where some of these issues are addressed (maybe in an unsatisfactory way but they are at least an attempt). Similarly, the statement on the limitations on Lines 55-57 is also overstated (see <https://doi.org/10.1016/j.res.2021.108184>).

Overall manuscripts says that there are limitations but does not really explain how this limitations come about.

2. Before going into the results (Section 1), it would be better, I think, to explain in more details the proposed approach and its theoretical merits.

3. TV is somewhat an unfortunate acronym, since it's hard not to think of television when reading it. It is also unnecessary. I would suggest just spelling out Tomorrowville.

4. Some of the work on Tomorrowville is similar to the one on Centerville (<https://doi.org/10.1080/23789689.2016.1254999>). It would be good to be more clear about the novelty and contributions.

In summary, I think this is a very interesting paper. My comments above are minor and intended to help the authors make their contributions more clear.

Reviewer #2

(Remarks to the Author)

In "Towards inclusive risk-informed infrastructure development in expanding", the authors present a novel approach for integrating socio-economic consequences (e.g. gentrification) into critical infrastructure development and demonstrate it with an example in the context of natural hazard occurrences. The proposed framework contains 2 parts: i) a mathematical formulation of network performance given a set of assumptions about the location of services and households, and ii) an agent-based model used to estimate gentrification impacts using economic utility functions. The case study consists of a hypothetical scenario with determined geographical characteristics and some assumptions in the form of parameters for both parts of the framework (i.e. The mathematical formulation and the agent-based model). The authors present statistics about the number of people suffering gentrification in both a scenario optimized exclusively for performance and a scenario in which the optimization process is constrained to a maximum level of gentrification (750 people).

I believe the authors addressed a very interesting topic. I have a few major comments to improve the clarity and identify the strengths of the manuscript.

In summary, I am worried it is not clear what the contribution of this work is. My suggestion is that this paper be reframed as either a results-oriented paper focused on residential location decision-making or a more methods-focused paper about the integration of infrastructure models and ABMs for addressing social consequences. What direction the authors choose is up to them, but in either case I believe the following comments will improve the analysis.

*In general, the analysis at present requires an extensive sensitivity analysis of the results. The effect of many variables in the problem are ignored, the most important of which in my opinion is geography, in particular, the specific locations of services/households/roads/economic activity about the zone at risk because of natural hazards. This would greatly influence the results since geography is the main driver of the risk and also of the performance metrics (generally described as driving distances) and thus I do not believe it is enough to test one hypothetical case study to justify the claims about the capabilities of the framework in a real-world scenario. There should at least be an extensive sensitivity analysis on the maximum gentrification allowed parameter. What are the changes in performance as we increase/decrease it? What is the marginal "cost" in performance for each household gentrification avoided?

*The dynamics emerging from the agent-based model are critical for understanding the results, yet they are not discussed or shown at all. What if wealthier people live closer to the natural hazard area? Given that the model uses a willingness to pay threshold, would that mean that less movement of people would occur? Please include experiments aimed at understanding the effect of including this ABM for this particular problem and how those results generalize to other instances and with different assumptions for the parameters.

*What conditions are required for gentrification to occur in a natural hazard situation? Under what scenarios it is advisable to use this framework? If low-income people inhabit the zone at risk of becoming disconnected, is it realistic to assume they would move? Is it realistic to assume that high-income households would move to risky areas? Is risk part of the perceived value of a house?

*The pro-poor vision is imposed into the optimization model without justification. It should at least be contrasted with a solution of the optimization problem that doesn't include either the pro-poor or the holistic (gentrification) perspective.

*The only exploration presented in the results is in terms of people without access, yet the optimization model is defined as minimizing traveling distances. How much is traveling distance optimized in both scenarios? (Please use interpretable numbers, the Z value is adequate for comparing experiments but it is not transparent with the actual implications of adding this constraint). If we care more about how many people are left without access (since the simulated hazard is a flood obstructing roads), how much it changes the model to use a binary performance metric indicating whether or not access from each household to each service is possible given the configuration of the network?

*The introduction needs to much more clearly explain the contribution of the paper and describe the methods used to support the results. The results and discussion sections are not complete enough to understand the work presented without the methodology. In particular, I believe the connection between the model and the real world problem is obscure up to the very end of the methods, thus it is impossible to assign any value to the results shown. A major revision of the writing of the introduction and the results is needed.

*The results are not generalizable unless a thorough exploration of the variables explaining the "performance" gap between the performance-oriented and the holistic solution is presented. Toward the end of generalization, how would this framework be applied to something like citing EV charging stations, locating resilience hubs, hardening power substations, building seawalls, or other 'typical' infrastructure decisions?

Minor comments

- The phrase "Forward-looking people-centered" in Line 51 is brought up several times in the text. I believe this is misleading since the framework is only designed to include gentrification effects.
- Since all the numbers in Figure 5 are positive, I suggest you add them as stacked bars in Figure 3 and label the respective percentages they represent next to the bars. This would make it easier to contextualize the difference between the two models (in other words, we could "see" the cost of avoiding gentrification of 168 people in number of people losing access)
- Line 153: How is SDG 10 addressed? As far as the current results show, the pro-poor nature of the optimization process is not part of the contribution since an experiment without it was not tested and it was not mentioned as an important part of the methods.

Reviewer #3

(Remarks to the Author)

The manuscript present risk-informed decision-making framework for urban infrastructure development which balances performance and socio-economic impact. The proposed framework has merits in the field of infrastructure risk management in general since it actively considers gentrification impact in infrastructure planning. However, the manuscript has some major issues that need to be addressed. I recommend accept with major revision.

1. Limited literature review.

- a. L46-48. The reviewer doesn't agree with this point. Most of infrastructure modeling techniques are general and adaptive so with simple input change, they are applicable to expanding infrastructure.
- b. L48-50. L54-57. The reviewer doesn't agree with these points. These points may be true for the cited articles. However,

there exist other works address these points already. See the recent works done on community resilience by the NIST Center of Excellence for Community resilience, NSF NHERI, etc. You can find more information at: He, X., & Cha, E. J. (2022). State of the research on disaster risk management of interdependent infrastructure systems for community resilience planning. *Sustainable and Resilient Infrastructure*, 7(5), 391-420.

c. Overall, the literature review of the current draft is very limited and draws conclusions too quickly before comprehensively reviewing the state-of-the-art research, which needs to be addressed to provide a proper motivation for this study.

2. Figures 3 & 4. These figures are supposed to be compared. However, the difference is very hard to read. The authors should find another way of presentation to highlight the comparison.

My recommendation is to accept with a major revision.

Version 1:

Reviewer comments:

Reviewer #2

(Remarks to the Author)

I really appreciate all the work the authors spent in responding to my review. Thank you for taking the comments seriously and for the tremendous amount of work you put in.

On balance, I am very comfortable with the changes made by the authors and the general form of the manuscript presently. I have one final comment prior to publication based on the review response documents:

Please integrate the extensive rebuttal you wrote for Q2.4 and Q2.5 in the review response document for reviewer 2 into the discussion of the manuscript. I am well convinced by the page-or-so response, but the few paragraphs which made it to the paper do not fully capture the nature of what you replied in the review rebuttal.

Towards inclusive risk-informed infrastructure development in expanding cities

We thank the reviewers for their very helpful feedback that significantly improved the manuscript. Their comments are listed in order below in italic text, followed by our responses in normal text. Excerpts from the revised text are written in blue. Any line numbers mentioned refer to the clean version of the revised manuscript.

Reviewer #1

Q1.1 The manuscript is interesting, timely, and overall well-written. I recommend publication after the following comments are addressed.

Reply: We thank the reviewer for the positive assessment of our work. We have made several changes in response to their comments, which are now discussed in detail.

Q1.2 The stated limitations of the current approaches (Lines 46-49) are somewhat unfairly representing the strengths of the literature and existing approaches. It's not clear why the existing approaches would not be suitable for future hazards, and the statement that they ignore the context/stakeholder priorities is at least too strong. See for example <https://doi.org/10.1080/23789689.2018.1448667>, <https://doi.org/10.1111/j.1539-6924.2006.00801.x>, and <https://doi.org/10.1061/AJRUA6.00009> where some of these issues are addressed (maybe in an unsatisfactory way but they are at least an attempt). Similarly, the statement on the limitations on Lines 55-57 is also overstated (see <https://doi.org/10.1016/j.res.2021.108184>).

Overall manuscripts says that there are limitations but does not really explain how this limitations come about.

Reply: The limitations of existing approaches that the reviewer refers to are: (a) their unsuitability for informing decision-making around the design of expanding infrastructure and its resilience to future hazards; (b) their failure to account for context-specific external stakeholder priorities (visions) around infrastructure performance; and (c) their neglect of any unintended, undesirable socioeconomic consequences (e.g., gentrification or population segregation) that accompany the optimization of infrastructure performance for hazard-induced impacts.

We agree with the reviewer's opinion on (a) and have removed the corresponding statement from the revised manuscript. We now acknowledge the existence of approaches for facilitating risk-sensitive design of infrastructure, through the following sentence (lines 47 to 51): "For instance, studies of natural-hazard-induced functionality losses to transportation systems have centred on travel delay, unmet demand, and loss of connectivity (e.g., Adey et al. 2004; Bell et al. 2008; Decò and Frangopol 2013; Dong et al. 2014; Hackl et al. 2018), which have been useful for informing their risk-sensitive design (Faturechi and Miller-Hooks 2014; Chang 2016; He and Cha 2022; Silva-lopez et al. 2022a)."

We agree that the statement made on (b) in the original version of the manuscript was too strong. However, we think it is important to acknowledge that existing related studies primarily focus on the top-down priorities of infrastructure owners and operators rather than the bottom-up needs of actual infrastructure users (unlike the proposed framework). The corresponding text in the revised manuscript (lines 58 to 61) now reads as follows: "Firstly, they typically overlook the context-specific bottom-up infrastructure performance needs of diverse users (including those from vulnerable communities) and focus instead on the top-

down perspectives of infrastructure owners and operators (e.g., Chang et al. 2014; McDaniels and Chang 2015).”

We do not believe that our comments on (c) are unfair or exaggerated. The study mentioned by the reviewer in relation to (c) examines the effects of infrastructure performance losses created by natural hazard events; it does not capture any *unintended*, undesirable socioeconomic consequences that are not directly related to disasters.

We thank the reviewer for providing additional literature that focuses on the societal consequences of hazard-induced disruption to transportation infrastructure. We have better captured the extent of existing work on these consequences in the revised manuscript, by:

1. Adding literature to support one of the first statements made in the introduction section, which now reads as: “However, the quality of critical infrastructure performance can be hampered by the occurrence of natural-hazard events (including those related to climate change), which can lead to significant societal impacts, including casualties, food insecurity, population displacement, business interruption, and unemployment (Chang 2016; Gardoni and Murphy 2020).”
2. Adding the following statement to the introduction section (in lines 52 to 56): “Further work on these systems has focused on transforming functionality losses into broader societal consequences of hazard-induced disruption, including accessibility impacts to essential services/locations like hospitals, schools, and shelters (e.g., Serulle and Cirillo 2014; Wein et al. 2014; Miller and Baker 2016) as well as effects on wellbeing experienced by people across different socioeconomic groups (e.g., Silva-lopez et al. 2022b; Boakye et al. 2022).”

Q1.3 Before going into the results (Section 1), it would be better, I think, to explain in more details the proposed approach and its theoretical merits.

Reply: We thank the reviewer for this valuable suggestion. We have addressed this comment by moving the methods section before the results section, such that the proposed approach and its theoretical merits are now discussed before the results are introduced.

Q1.4 TV is somewhat an unfortunate acronym, since it's hard not to think of television when reading it. It is also unnecessary. I would suggest just spelling out Tomorrowville.

Reply: We understand the reviewer’s concern. We have accordingly removed the “TV” acronym and instead use the full “Tomorrowville” name. Note that we have kept the term “TV0”, which refers to a specific configuration of Tomorrowville, to maintain consistency with previous literature that introduced and/or used the Tomorrowville testbed – see: <https://doi.org/10.1016/j.ijdr.2023.103651>, <https://doi.org/10.1016/j.ijdr.2022.103365>, <https://doi.org/10.1016/j.ijdr.2022.103400>

Q1.5 Some of the work on Tomorrowville is similar to the one on Centerville (<https://doi.org/10.1080/23789689.2016.1254999>). It would be good to be more clear about the novelty and contributions.

Reply: As mentioned in response to the previous comment, Tomorrowville has already been used extensively in previous literature. Therefore, we feel that the current manuscript would not be a suitable venue for a discussion on the novelty and contributions of Tomorrowville.

We emphasise that Tomorrowville is a completely separate entity to Centerville that was developed for a significantly different (i.e., Global South) context, and (unlike Centerville)

contains detailed information on individual people and households in addition to physical infrastructure; the similarities between the two testbeds are extremely limited.

Q1.6 In summary, I think this is a very interesting paper. My comments above are minor and intended to help the authors make their contributions more clear.

Reply: Once again, we thank the reviewer for their important constructive feedback. We agree with the reviewer that the resulting modifications have enhanced the clarity of our contributions.

Reviewer #2

Q2.1 I believe the authors addressed a very interesting topic. I have a few major comments to improve the clarity and identify the strengths of the manuscript.

Reply: We thank the reviewer for their detailed, insightful comments that have greatly enhanced the quality of the manuscript. These comments are now addressed one by one.

Q2.2 In summary, I am worried it is not clear what the contribution of this work is. My suggestion is that this paper be reframed as either a results-oriented paper focused on residential location decision-making or a more methods-focused paper about the integration of infrastructure models and ABMs for addressing social consequences. What direction the authors choose is up to them, but in either case I believe the following comments will improve the analysis.

Reply: We thank the reviewer for this valuable suggestion. In line with this comment, we have reframed the paper as a methods-focused one, with the case study application simply intended to provide a demonstration of the proposed methodology. We have done this by moving the methods section up before the results section, such that the proposed approach and its theoretical merits are now discussed as a main focus of the paper, before the results are introduced. Further substantial changes that have been made to create a more methods-focused manuscript are detailed in response to specific subsequent comments by the reviewer.

Q2.3 In general, the analysis at present requires an extensive sensitivity analysis of the results. The effect of many variables in the problem are ignored, the most important of which in my opinion is geography, in particular, the specific locations of services/households/roads/economic activity about the zone at risk because of natural hazards. This would greatly influence the results since geography is the main driver of the risk and also of the performance metrics (generally described as driving distances) and thus I do not believe is it enough to test one hypothetical case study to justify the claims about the capabilities of the framework in a real-world scenario. There should at least be an extensive sensitivity analysis on the maximum gentrification allowed parameter. What are the changes in performance as we increase/decrease it? What is the marginal "cost" in performance for each household gentrification avoided?

Reply: We appreciate this comment. As mentioned in response to the previous comment, we have made the central focus of our paper its methodological contributions, and the case study application is simply intended to provide a demonstration of these. As such, we believe that one case study is sufficient, in line with many other methods-focused papers that have been published in this journal, e.g., :

1. <https://doi.org/10.1038/s44172-024-00162-y>
2. <https://doi.org/10.1038/s44172-023-00079-y>
3. <https://doi.org/10.1038/s44172-024-00258-5>

However, we agree with the reviewer on the need to consider sensitivity analyses. In line with the reviewer's suggestion, we have explored the relationship between gentrification (i.e., the number of expected triggered relocations, $E(\varepsilon)$) and infrastructure performance (in terms of Z) for the 20 Tomorrowville road expansion possibilities with the highest Z that comply with all constraints of the optimisation except (possibly) $\varepsilon \leq \varepsilon_t$. These results, which are presented and discussed in lines 739 to 757 of the Appendix, reveal an unclear relationship between Z and ε . This is because higher values of Z lead to increases in both a buyer's willingness to pay ($WTP_{r,b}$) and the prices set by the seller ($P_{r,s}$), and it is the trade-off between the two that determine the value of ε for a given infrastructure development proposal. In summary, it is not always the case that better infrastructure performance entails an additional gentrification cost; however, explicitly accounting for both variables enables their complicated relationship to be tracked, ultimately facilitating more informed, responsible decision-making.

The sensitivity analyses are referred to in the main text at the end of the results section (in lines 418 to 420), as follows: “Note that further investigation of the relationship between prices, infrastructure performance, and triggered relocations can be found in the Appendix.” We also refer to this sensitivity analysis in the discussion section (lines 448 to 451), as follows: “Sensitivity analyses involving some of the case study parameters confirmed a complicated relationship between infrastructure performance and gentrification; better infrastructure performance does not necessarily entail an additional gentrification cost, underlining the importance of explicitly tracking both variables for informed decision making.”

Furthermore, we have added some cautious language to the discussion, to emphasise that the conclusions drawn from our work are based on just one case study application, and further testing of the framework for real-world applications is required. This text (in lines 454 to 458) reads as follows: “It is important to note that these conclusions are based on one example involving the road infrastructure of a Global South virtual testbed in the presence of flooding, where there is a deliberately significant spatial correlation between income and level of hazard exposure (Agrawal et al. 2024). Further testing of the framework’s capabilities in real-world settings is required...”

Q2.4 The dynamics emerging from the agent-based model are critical for understanding the results, yet they are not discussed or shown at all. What if wealthier people live closer to the natural hazard area? Given that the model uses a willingness to pay threshold, would that mean that less movement of people would occur? Please include experiments aimed at understanding the effect of including this ABM for this particular problem and how those results generalize to other instances and with different assumptions for the parameters.

Reply: Thanks for these questions.

Firstly, we have modified the text of Section 1.2 to better explain the meaning of the parameters that underly the ABM equations. We have clarified that:

- Utility relates to an “agent’s attractiveness to” a residential unit
- The $u(\lambda_{r,j})$ variable underlying the calculation of utility “is an objective representation of the benefit associated with the j^{th} attribute”, whereas the corresponding $\alpha_{i,j}$ variable is the “weight representing the subjective preference of the i^{th} agent towards attribute $\lambda_{r,j}$ ”
- Higher utility values “will yield a higher willingness to pay”
- The rental price set by a seller is “based on the benefit of the unit to the seller $U_{r,s}$ ”

It is important to note that willingness to pay does not account for natural hazard risk in the case-study application. Instead, risk, which is expressed in terms of accessibility losses and increases in distance travelled due to the occurrence of a natural hazard, is considered through the infrastructure performance component of the framework. The utility values that positively influence willingness to pay (as set out in equation 3 and explained in the corresponding text) account for “the road distance from each household’s residential unit to hospitals $\lambda_{r,1}$, schools $\lambda_{r,2}$, and workplaces $\lambda_{r,3}$ ”. Thus, the closer a residence is to hospitals, schools, and a given household’s workplace (in terms of road distance during normal operations), the higher the household’s willingness to pay for it.

We have better clarified the absence of risk from the willingness to pay calculation in Section 1.4 of the revised manuscript, where the $\lambda_{r,j}$ values that underly utility and willingness to pay are now described as follows (lines 333 to 335): “ $\lambda_{r,j}$ comprise the road distance (assuming “normal” day-to-day infrastructure performance) from each household’s

residential unit to hospitals $\lambda_{r,1}$, schools $\lambda_{r,2}$, and workplaces $\lambda_{r,3}$ ” We have also noted this as a limitation in the discussion section, as follows (lines 471 to 474): “Second, the case study application relies on hypothetical stakeholder inputs, assuming that infrastructure needs are limited to accessing hospitals, schools, and workplaces, and that a household’s attractiveness to a given residence is only determined by the travel distance to these critical infrastructure under normal conditions (i.e., when there are no disruptions from hazards). ”

We have introduced a set of experiments aimed at understanding the relationship between prices set by the seller, $P_{r,s}$, and the number of resulting triggered relocations (i.e., the number of occurrences of the price exceeding the willingness to pay of the buyer $P_{r,s} > WTP_{r,b}$) disaggregated by income (see lines 761 to 774 of the Appendix). The results of these experiments reveal that, for price increases up to 10% beyond those originally set in the case study (per the ABM seller parameters quantified in Section 1.4):

- There are no expected relocations of high-income households, whose available budget is sufficient to mean that their willingness to pay still exceeds the prices set
- Expected triggered relocations for middle-income households start when the price increases by approximately 6%
- Expected triggered relocations for low-income households grow exponentially with percent increase in price, indicating that these households have willingness to pay values that are very close to the original prices set by the seller

In summary, these experiments confirm that low-income households would remain particularly susceptible to triggered relocations if rental prices increased somewhat. The findings are now referred to in the discussion section of the revised manuscript (lines 452 to 454), as follows: “...these analyses revealed that low-income households would remain disproportionately susceptible to triggered relocations (and therefore would still benefit the most from a gentrification cap) even if rental prices increased somewhat (without considering any infrastructure expansion).”

Furthermore, it is important to note that the sensitivity analyses presented in lines 739 to 757 of the Appendix represent an additional set of experiments that help to increase understanding of the ABM. They reveal a complicated relationship between infrastructure performance, Z , and gentrification, ε . This is because infrastructure developments with bigger Z produce larger utility values for affected residences (by increasing their connectivity to schools, workplaces, and hospitals), which drive increases in both $P_{r,s}$ and $WTP_{r,b}$, and it is the trade-off between these increases that determine ε . The fact that the relocations occur exclusively to low-income households can be somewhat explained by their heightened susceptibility to evictions due to price increases (and the general similarities in values of corresponding $P_{r,s}$ and $WTP_{r,b}$; see previous paragraph), but also underlines the significant spatial correlation that exists between income and level of hazard exposure in Tomorrowville (Agrawal et al. 2024). The infrastructure performance optimisation process will pay specific attention to increasing the connectivity of (and therefore driving price increases in) hazard-prone areas, given its risk mitigation goal expressed through Z_2 . (The pro-poor nature of the optimisation process is another reason why infrastructure development is predominantly focused in low-income areas.)

If higher income households resided in hazard-prone areas instead, the number of triggered relocations would change depending on the trade-off between values of $P_{r,s}$ and $WTP_{r,b}$ (which would be larger than those associated with low-income households). We do not investigate this hypothetical scenario, because the strong spatial relationship between income and hazard exposure that exists within Tomorrowville has been specifically designed

to mirror real-world contexts (see Agrawal et al. 2024). Therefore, reconfiguring Tomorrowville by placing high-income households in hazard-prone areas would create a somewhat unrealistic experiment. We acknowledge the significant relationship between income and hazard exposure in Tomorrowville in the discussion section of the revised manuscript, as follows: “It is important to note that these conclusions are based on one example involving the road infrastructure of a Global South virtual testbed in the presence of flooding, where there is a deliberately significant spatial correlation between income and level of hazard exposure (Agrawal et al. 2024).”

Q2.5 What conditions are required for gentrification to occur in a natural hazard situation? Under what scenarios it is advisable to use this framework? If low-income people inhabit the zone at risk of becoming disconnected, is it realistic to assume they would move? Is it realistic to assume that high-income households would move to risky areas? Is risk part of the perceived value of a house?

As mentioned in response to the previous comment, gentrification is a direct result of price increases driven by infrastructure development, rather than the occurrence of a natural hazard event. Risk is not considered part of the utility function used in the case study application to determine willingness to pay (and therefore the perceived value of a house); we have discussed and addressed this limitation in response to the previous comment.

It is realistic to assume that residences with greater connectivity to important locations are valued higher (e.g., <https://link.springer.com/article/10.1007/s11116-011-9379-0>; <https://doi.org/10.3141/1994-09>; <https://www.sciencedirect.com/science/article/pii/0094119088900356>). Therefore, if infrastructure development takes place in hazard-prone areas, it is realistic to assume that house prices will increase there such that low-income households may be forced out of their homes. We recognise that it is unrealistic to assume that these low-income households would be replaced with high-income households, and this could be avoided by forcing household movements to take place within nearby neighbourhoods only. (Note that, in any case, middle-income or more well-off low-income households may well be more attracted to these hazard-prone residences than high-income ones based on the current utility functions used, depending on the locations of their workplaces). However, tracking household movements beyond triggered relocations is outside the scope of this study.

Finally, it is important to note that the infrastructure development process helps to mitigate loss of accessibility due to natural hazards, and therefore anyway reduces the riskiness of “risky areas”.

Q2.6 The pro-poor vision is imposed into the optimization model without justification. It should at least be contrasted with a solution of the optimization problem that doesn't include either the pro-poor or the holistic (gentrification) perspective.

Reply: Many thanks for this comment. The pro-poor vision integrated into the optimisation model reflects an increasing emphasis in the literature on the need to integrate pro-poor thinking and approaches in disaster risk management and assessment (e.g., <https://doi.org/10.12962/j2355262x.v11i1.a506>; <https://doi.org/10.1080/00396265.2016.1212160>), which is also recognised in the guiding principles of the Sendai Framework for Disaster Risk Reduction (<https://www.undrr.org/media/16176/download?startDownload=20241028>). We have clarified this point in the text where the pro-poor perspective of the hypothetical case study stakeholders is introduced (lines 314 to 322, Section 1.4), as follows: “The hypothetical stakeholders are assumed to.... hold a pro-poor vision on future urban expansion, in line

with the latest thinking on disaster risk management and assessment (Galasso et al. 2021; Agrawal et al. 2024) as well as the guiding principles of the Sendai Framework for Disaster Risk Reduction (UNDRR, 2015).”

Given the importance of adopting a pro-poor perspective as established above, we have decided not to include any analysis in our case study demonstration that is not pro-poor. However, it is important to note that we do compare the performance-oriented and holistic solutions in terms of pro-poorness. The holistic infrastructure expansion leads to a better pro-poor outcome than the performance-oriented one, resulting in 24.3% less triggered relocations among low-income households while only increasing their accessibility losses by 5%. This finding is discussed in the results section (lines 409 to 418): “The holistic infrastructure expansion reduces $E(\varepsilon)$ and Z by 24.3% and 3.8%, respectively, compared to the performance-oriented expansion.... The reduction in Z translates into a minor rise (~5%) in post-flood accessibility losses for low-income households.... These observations specifically highlight the value of adopting a well-rounded perspective in a deliberately pro-poor development process, i.e., gaining a significant reduction in the number of evicted low-income households at the expense of accepting noticeable lower infrastructure performance for those more well off.” Furthermore, we clarify in the discussion section (lines 445 to 448) that: “The case study demonstrated that the framework could produce an infrastructure expansion with tolerable unintended consequences (specifically benefitting low-income households) at the expense of only slightly sub-optimal performance (predominantly impacting mid- and high-income households).”

Q2.7 The only exploration presented in the results is in terms of people without access, yet the optimization model is defined as minimizing traveling distances. How much is traveling distance optimized in both scenarios? (Please use interpretable numbers, the Z value is adequate for comparing experiments but it is not transparent with the actual implications of adding this constraint). If we care more about how many people are left without access (since the simulated hazard is a flood obstructing roads), how much it changes the model to use a binary performance metric indicating whether or not access from each household to each service is possible given the configuration of the network?

Reply: We appreciate this comment, but have ultimately elected to keep our results in terms of Z for the following reasons:

1. Z represents the objective function, and therefore seems a reasonable variable to discuss when referring to the results of the optimisation process.
2. Z summarises information on infrastructure performance in different time periods. In the case-study application, these periods represent (1) normal day-to-day times when a hazard has not occurred and (2) the immediate post-hazard response period (the longer recovery-term phase is neglected since $\gamma_3 = 0$; see Section 1.4). The reviewer’s suggestion would mean reporting multiple optimised travel distances for each scenario, which would complicate the discussion of the results.
3. Z accounts for travel distances to multiple locations, corresponding to the number of infrastructure needs of the household. Thus, the reviewer’s suggestion would also mean reporting multiple sets of travel distances per considered time period and scenario, further complicating the discussion of the results.

We believe that our additional exploration of the relationship between Z and ε (see lines 739 to 757 of the Appendix and our responses to Q2.3 and Q2.4) helps to shed light on the implications of constraining infrastructure performance in line with a pre-defined restriction on gentrification, without the need to investigate multiple types of travel distances associated with different time periods.

It is important to note that we do report the number of people who have lost access to different locations, for both the performance-oriented and holistic solutions. This is a binary performance metric; it tracks the number of times households are disconnected from locations of interest (i.e., when their travel distance goes to infinity). Loss of accessibility is reported in lines 386 to 391 of the results section. We have slightly modified our reporting of these results in the revised version of the manuscript, to clarify what loss of accessibility means in terms of travel distance. The revised statement of these results reads as follows: “... which translates into a post-flood loss of connectivity (caused by impassable roads creating a theoretical travel distance of infinity) to hospitals, schools, and workplaces for, on average, 1,684, 1,414, and 3,722 households, respectively”. We do not optimise based on the loss of accessibility metric, since this would limit us to only considering infrastructure performance in the post-hazard period; instead, we conduct a broader investigation that also accounts for infrastructure performance during “normal” everyday times.

Q2.8 The introduction needs to much more clearly explain the contribution of the paper and describe the methods used to support the results. The results and discussion sections are not complete enough to understand the work presented without the methodology. In particular, I believe the connection between the model and the real world problem is obscure up to the very end of the methods, thus it is impossible to assign any value to the results shown. A major revision of the writing of the introduction and the results is needed.

Reply: We appreciate this valuable comment and have addressed it by:

1. Moving the methods section up before the results section, such that the proposed approach and its theoretical merits are now discussed as a main focus of the paper, before the results are introduced (as mentioned previously).
2. Significantly modifying the the introduction to provide a more comprehensive overview of relevant existing studies (especially those that attempt to assess societal consequences of hazard-induced disruption to transportation systems) and to more clearly highlight the contributions of this study, which can be summarized as follows:
 - Considering the bottom-up needs of infrastructure users, rather than only the top-down priorities of infrastructure owners and operators. This contribution (herein referred to as novelty #1) facilitates a unique, people-centered approach to decision-making on risk-informed infrastructure development.
 - Accounting for the unintended, undesirable socioeconomic consequences (e.g., gentrification or population segregation) that might accompany the optimization of infrastructure performance for hazard-induced impacts (herein referred to as novelty #2).

The corresponding modifications to the introduction are now summarized.

- We have added literature to support one of the first statements made in the introduction section on the societal consequences of infrastructure disruption. This statement now reads as follows (lines 30 to 34): “However, the quality of critical infrastructure performance can be hampered by the occurrence of natural-hazard events (including those related to climate change), which can lead to significant societal impacts, including casualties, food insecurity, population displacement, business interruption, and unemployment (Chang 2016; Gardoni and Murphy 2020).”

- The following statement on the societal consequences of hazard-induced disruptions to transportation systems has been added (lines 52 to 56): “Further work on these systems has focused on transforming functionality losses into broader societal consequences of hazard-induced disruption, including accessibility impacts to essential services/locations like hospitals, schools, and shelters (e.g., Serulle and Cirillo 2014; Wein et al. 2014; Miller and Baker 2016) as well as effects on wellbeing experienced by people across different socioeconomic groups (e.g., Silva-lopez et al. 2022b; Boakye et al. 2022).”
- To better emphasise novelty #1 and novelty #2, the corresponding limitations of existing literature are now mentioned together in the introduction section, as follows (lines 58 to 74): “Firstly, they typically overlook the context-specific bottom-up infrastructure performance needs of diverse users (including those from vulnerable communities) and focus instead on the top-down perspectives of infrastructure owners and operators (e.g., Chang et al. 2014; McDaniels and Chang 2015). This is a crucial shortcoming, given that people-centered infrastructure risk management approaches are becoming progressively more important in the context of climate change, rapid population growth, and increasingly interconnected urbanization (Cremen et al. 2022b; Filippi et al. 2023). Furthermore, current infrastructure risk modeling approaches/models do not capture any unintended, undesirable socioeconomic consequences of optimizing infrastructure performance for hazard-induced impacts, such as gentrification or population segregation (Bagheri-Jebelli et al. 2021) that are not uncommon. For instance, improved connectivity that accompanied the development of China’s expansive high-speed rail network resulted in rapid urbanization that unintentionally led to the loss of agricultural land (Yu et al. 2022) through haphazard urban sprawl (Zhu 2021). Investments in bus and train systems within South Africa’s Gauteng province, which aimed to create efficient and accessible urban transit, have reinforced spatial segregation (Musakwa and Gumbo 2017). These examples underscore the critical need to address and/or mitigate unintended socioeconomic consequences in infrastructure planning and expansion.”
- We now acknowledge the fact that our framework is not unique in its ability to facilitate risk-informed infrastructure design. We have removed the following statement from the revised version of the manuscript: “Firstly, they are developed for application to current infrastructure with static and often deterministic representations of exposure and vulnerability, so they fall short in facilitating decision-making around expanding infrastructure for resilience to future hazards (Cremen et al. 2022a).” Furthermore, we have added existing literature on the risk-sensitive design of infrastructure, in the following sentence (lines 47 to 52): “For instance, studies of natural-hazard-induced functionality losses to transportation systems have centred on travel delay, unmet demand, and loss of connectivity (e.g., Adey et al. 2004; Bell et al. 2008; Decò and Frangopol 2013; Dong et al. 2014; Hackl et al. 2018), which have been useful for informing their risk-

sensitive design (Faturechi and Miller-Hooks 2014; Chang 2016; He and Cha 2022; Silva-lopez et al. 2022a).”

3. Improving the explanation of the ABM, as detailed in our response to Q2.4.
4. Adding the sensitivity analyses, as detailed in our response to Q2.3 and Q2.4.
5. Significantly modifying the discussion section, such that it now:
 - More accurately highlights the contribution of the work. Reference to its focus on future infrastructure expansion has been removed, in line with the modifications made to the introduction section. The novelty statement (lines 424 to 428) now reads as follows: “The framework extends beyond conventional natural hazard infrastructure impact assessments by facilitating external participation in the design process and holistically accounting for unintended consequences of risk-informed infrastructure development (e.g., gentrification), recognizing that equitable development may ultimately require a departure from a strictly performance-driven outlook.”
 - More clearly explains how the framework addresses SDG 10 on reducing inequalities; see our response to Q2.1.3.
 - Discusses the significance of the sensitivity analyses results (in lines 448 to 454), as follows: “Sensitivity analyses involving some of the case study parameters confirmed a complicated relationship between infrastructure performance and gentrification; better infrastructure performance does not necessarily entail an additional gentrification cost, underlining the importance of explicitly tracking both variables for informed decision making. Furthermore, these analyses revealed that low-income households would remain disproportionately susceptible to triggered relocations (and therefore would still benefit the most from a gentrification cap) even if rental prices increased somewhat (not considering any infrastructure expansion).”
 - More clearly explains the limitations of the study, through the following statements that have been added: (1; lines 454 to 458) “It is important to note that these conclusions are based on one example involving the road infrastructure of a Global South virtual testbed in the presence of flooding, where there is a deliberately significant spatial correlation between income and level of hazard exposure (Agrawal et al. 2024). Further testing of the framework’s capabilities in real-world settings is required...” (2; lines 471 to 474) “... the case study application relies on hypothetical stakeholder inputs, assuming that.... a household’s attractiveness to a given residence is only determined by the travel distance to these critical infrastructure under normal conditions (i.e., when there are no disruptions from hazards).”
 - Clearly states that the proposed framework could be generalised to decision-making on other infrastructure development, as follows (lines 458 to 462): “...the proposed formulation is general enough for application to any hazardscape and critical infrastructure challenge of interest (e.g., from siting electric vehicle chargers to constructing seawalls) involving a trade-off between enhancing the performance of the infrastructure and limiting triggered relocations (i.e., gentrification) caused by resulting price increases.”

Q2.9 The results are not generalizable unless a thorough exploration of the variables explaining the "performance" gap between the performance-oriented and the holistic solution

is presented. Toward the end of generalization, how would this framework be applied to something like citing EV charging stations, locating resilience hubs, hardening power substations, building seawalls, or other 'typical' infrastructure decisions?

Reply:

The relationship between gentrification (i.e., the number of triggered relocations, ε) and infrastructure performance (in terms of Z) explored in the Appendix (and described in response to previous comments) sheds light on the performance gap between the performance-oriented and holistic solutions. In summary, higher values of Z - caused by greater connectivity to critical locations - lead to increases in both a buyer's willingness to pay ($WTP_{r,b}$) and the prices set by the seller ($P_{r,s}$), and it is the trade-off between the two that determine the value of $E(\varepsilon)$ for a given infrastructure development proposal.

The framework deals with any infrastructure challenge where there is a trade-off between enhancing the performance of the infrastructure and minimising triggered relocations (i.e., gentrification) caused by resulting price increases. Thus, the framework could be used for any of the infrastructure decisions mentioned by the reviewer, so long as the aspect of infrastructure performance to be optimised is reflected in the utility functions of the gentrification ABM. The discussion section now includes a longer explanation of how the framework can be generalised, which reads as follows (lines 458 to 462): “.. the proposed formulation is general enough for application to any hazardscape and critical infrastructure challenge of interest (e.g., from siting electric vehicle chargers to constructing seawalls) involving a trade-off between enhancing the performance of the infrastructure and limiting triggered relocations (i.e., gentrification) caused by resulting price increases.”

Minor comments (Reviewer #2)

Q2.1.1 The phrase "Forward-looking people-centered" in Line 51 is brought up several times in the text. I believe this is misleading since the framework is only designed to include gentrification effects.

Reply: Thanks for this comment. The term “forward-looking people-centred” appeared in the original text once, within the introduction section. We have removed this term from the revised manuscript, as part of the changes made to the literature review (which are outlined in response to previous comments). As a result, the phrase “forward-looking” no longer features in the manuscript. The term “people-centred” refers to the prioritisation of the needs and concerns of specific groups or communities (see here, for instance). We clarify how the proposed framework is people-centred in the in the second-last paragraph of the revised introduction section, as follows (lines 77 to 78): “The framework integrates an optimization procedure for expanding infrastructure that (i) balances its performance (in terms of bespoke user needs)...”

Q2.1.2 Since all the numbers in Figure 5 are positive, I suggest you add them as stacked bars in Figure 3 and label the respective percentages they represent next to the bars. This would make it easier to contextualize the difference between the two models (in other words, we could "see" the cost of avoiding gentrification of 168 people in number of people losing access)

Reply: Thanks for this comment. We have addressed it by modifying Figure 3, such that it now also displays the number of households who lose accessibility in the case of the holistic expansion (see Figure A). The additional information now shown in Figure 3 has rendered Figure 5 redundant, so it has been removed from the revised manuscript.

Figure A (equivalent to Figure 3 of the revised manuscript). Number of households with no access to hospitals, schools, and workplaces for the existing road infrastructure in Tomorrowville (baseline), the performance-oriented expansion, and the holistic expansion, disaggregated by income level.

Q2.1.3 Line 153: How is SDG 10 addressed? As far as the current results show, the pro-poor nature of the optimization process is not part of the contribution since an experiment without it was not tested and it was not mentioned as an important part of the methods.

Reply:

Thanks for the question. SDG10 (“Reduce inequality within and among countries”) is addressed through the participatory nature of the proposed framework, which can help to prioritize the needs of lower-income populations and generally support a pro-poor approach in future risk-informed urban development. We have clarified this in the discussion section (lines 437 to 444), as follows: “The holistic, inherently participatory nature of the proposed framework can help to prioritize the needs of lower-income populations and generally support a pro-poor approach in future risk-informed urban development (e.g., Galasso et al. 2021). As such, the framework is cross-cutting, addressing broad sustainable development goals (i.e., SDG 10: *Reduced inequalities*) as well those that are more specifically focused on engineered assets (i.e., SDG 9: *Build resilient infrastructure, promote inclusive and sustainable industrialization and foster innovation*; SDG 11: *Make cities and human settlements inclusive, safe, resilient and sustainable*).

Reviewer #3

Q3.1 The manuscript present risk-informed decision-making framework for urban infrastructure development which balances performance and socio-economic impact. The proposed framework has merits in the field of infrastructure risk management in general since it actively considers gentrification impact in infrastructure planning. However, the manuscript has some major issues that need to be addressed. I recommend accept with major revision.

Reply: We acknowledge the reviewer for their generally positive assessment of our manuscript. Their helpful comments, which have improved the quality of the manuscript, are now addressed in detail.

Q3.1 Limited literature review.

Q3.1a L46-48. The reviewer doesn't agree with this point. Most of infrastructure modeling techniques are general and adaptive so with simple input change, they are applicable to expanding infrastructure.

Reply: We agree with the reviewer and have removed the statement from the revised manuscript. We now acknowledge the existence of approaches for facilitating risk-sensitive design of infrastructure, through the following sentence (lines 47 to 52): "For instance, studies of natural-hazard-induced functionality losses to transportation systems have centred on travel delay, unmet demand, and loss of connectivity (e.g., Adey et al. 2004; Bell et al. 2008; Decò and Frangopol 2013; Dong et al. 2014; Hackl et al. 2018), which have been

useful for informing their risk-sensitive design (Faturechi and Miller-Hooks 2014; Chang 2016; He and Cha 2022; Silva-lopez et al. 2022a).”

Q3.1b L48-50. L54-57. The reviewer doesn't agree with these points. These points may be true for the cited articles. However, there exist other works address these points already. See the recent works done on community resilience by the NIST Center of Excellence for Community resilience, NSF NHERI, etc. You can find more information at: He, X., & Cha, E. J. (2022). State of the research on disaster risk management of interdependent infrastructure systems for community resilience planning. Sustainable and Resilient Infrastructure, 7(5), 391-420.

Reply: We agree with the reviewer that the statement made in lines 48-50 of the original manuscript was too strong. However, we think it is important to acknowledge that existing related studies primarily focus on the top-down priorities of infrastructure owners and operators rather than the bottom-up needs of actual infrastructure users (unlike the proposed framework). The corresponding text in the revised manuscript (lines 58 to 61) now reads as follows: “*Firstly, they typically overlook the context-specific bottom-up infrastructure performance needs of diverse users (including those from vulnerable communities) and focus instead on the top-down perspectives of infrastructure owners and operators (e.g., Chang et al. 2014; McDaniels and Chang 2015).*”

We do not believe that our comments in lines 54-57 of the original manuscript were inaccurate. To the best of our knowledge, the proposed framework is the first in the field of disaster risk and resilience to capture *unintended*, undesirable socioeconomic consequences of infrastructure development that are not directly related to the occurrence of natural hazards. The He and Cha (2022) study mentioned by the reviewer is indeed an excellent and comprehensive review of existing approaches for modelling resilient infrastructure; it does not appear to mention any study that captures impacts beyond those caused by disasters.

Q3.1c Overall, the literature review of the current draft is very limited and draws conclusions too quickly before comprehensively reviewing the state-of-the-art research, which needs to be addressed to provide a proper motivation for this study.

Reply: We have significantly modified the introduction to provide a more comprehensive overview of relevant existing studies (especially those that attempt to assess societal consequences of hazard-induced disruption to transportation systems) and to more clearly highlight the contributions of this study, which can be summarized as follows:

1. Considering the bottom-up needs of infrastructure users, rather than only the top-down priorities of infrastructure owners and operators. This contribution (herein referred to as novelty #1) facilitates a unique, people-centered approach to decision-making on risk-informed infrastructure development.
2. Accounting for the unintended, undesirable socioeconomic consequences (e.g., gentrification or population segregation) that might accompany the optimization of infrastructure performance for hazard-induced impacts (herein referred to as novelty #2).

The corresponding modifications to the introduction are now summarized.

1. We have added literature to support one of the first statements made in the introduction section on the societal consequences of infrastructure disruption. This statement now reads as follows (lines 30 to 34): “*However, the quality of critical*

infrastructure performance can be hampered by the occurrence of natural-hazard events (including those related to climate change), which can lead to significant societal impacts, including casualties, food insecurity, population displacement, business interruption, and unemployment (Chang 2016; Gardoni and Murphy 2020).”

2. The following statement on the societal consequences of hazard-induced disruptions to transportation systems has been added (in lines 52 to 56): “Further work on these systems has focused on transforming functionality losses into broader societal consequences of hazard-induced disruption, including accessibility impacts to essential services/locations like hospitals, schools, and shelters (e.g., Serulle and Cirillo 2014; Wein et al. 2014; Miller and Baker 2016) as well as effects on wellbeing experienced by people across different socioeconomic groups (e.g., Silva-lopez et al. 2022b; Boakye et al. 2022).”
3. To better emphasise novelty #1 and novelty #2, the corresponding limitations of existing literature are now mentioned together in the introduction section (lines 58 to 74), as follows: “Firstly, they typically overlook the context-specific bottom-up infrastructure performance needs of diverse users (including those from vulnerable communities) and focus instead on the top-down perspectives of infrastructure owners and operators (e.g., Chang et al. 2014; McDaniels and Chang 2015). This is a crucial shortcoming, given that people-centered infrastructure risk management approaches are becoming progressively more important in the context of climate change, rapid population growth, and increasingly interconnected urbanization (Cremen et al. 2022b; Filippi et al. 2023). Furthermore, current infrastructure risk modeling approaches/models do not capture any unintended, undesirable socioeconomic consequences of optimizing infrastructure performance for hazard-induced impacts, such as gentrification or population segregation (Bagheri-Jebelli et al. 2021) that are not uncommon. For instance, improved connectivity that accompanied the development of China’s expansive high-speed rail network resulted in rapid urbanization that unintentionally led to the loss of agricultural land (Yu et al. 2022) through haphazard urban sprawl (Zhu 2021). Investments in bus and train systems within South Africa’s Gauteng province, which aimed to create efficient and accessible urban transit, have reinforced spatial segregation (Musakwa and Gumbo 2017). These examples underscore the critical need to address and/or mitigate unintended socioeconomic consequences in infrastructure planning and expansion.”
4. We now acknowledge the fact that our framework is not unique in its ability to facilitate risk-informed infrastructure design. We have removed the following statement from the revised version of the manuscript: “Firstly, they are developed for application to current infrastructure with static and often deterministic representations of exposure and vulnerability, so they fall short in facilitating decision-making around expanding infrastructure for resilience to future hazards (Cremen et al. 2022a). Furthermore, we have added existing literature on the risk-sensitive design of infrastructure, in the following sentence (lines 47 to 52): “For instance, studies of natural-hazard-induced functionality losses to transportation systems have centred on travel delay, unmet demand, and loss of connectivity (e.g., Adey et al. 2004; Bell et al. 2008; Decò and Frangopol 2013; Dong et al. 2014; Hackl et al. 2018), which have been useful for informing their risk-sensitive design (Faturechi and Miller-Hooks 2014; Chang 2016; He and Cha 2022; Silva-lopez et al. 2022a).”

Q3.2 Figures 3 & 4. These figures are supposed to be compared. However, the difference is very hard to read. The authors should find another way of presentation to highlight the comparison.

Reply: Thanks for this valuable feedback. It appears that the reviewer may be referring to Figures 2 and 4 (rather than Figures 3 and 4), which are directly comparable. To address this comment and better highlight the differences between both infrastructure expansions, we have placed both figures side by side as subpanels of one figure (now Figure 2). Corresponding modifications to the text that refer to these and other figures have also been made.

References

- Adey, B., Hajdin, R., & Brühwiler, E. (2004). Effect of common cause failures on indirect costs. *Journal of Bridge Engineering*, 9(2), 200-208.
- Agrawal, H., Wang, C., Cremen, G., & McCloskey, J. (2024). A geophysics-informed pro-poor approach to earthquake risk management. *Natural Hazards*, 1-19.
- Bell, M. G., Kanturska, U., Schmöcker, J. D., & Fonzone, A. (2008). Attacker–defender models and road network vulnerability. *Philosophical Transactions of the Royal Society A: Mathematical, Physical and Engineering Sciences*, 366(1872), 1893-1906.
- Boakye, J., Guidotti, R., Gardoni, P., & Murphy, C. (2022). The role of transportation infrastructure on the impact of natural hazards on communities. *Reliability Engineering & System Safety*, 219, 108184.
- Chang, S. E., McDaniels, T., Fox, J., Dhariwal, R., & Longstaff, H. (2014). Toward disaster-resilient cities: Characterizing resilience of infrastructure systems with expert judgments. *Risk analysis*, 34(3), 416-434.
- Chang, S. E. (2016). Socioeconomic impacts of infrastructure disruptions. In *Oxford research encyclopedia of natural hazard science*.
- Decò, A., & Frangopol, D. M. (2013). Life-cycle risk assessment of spatially distributed aging bridges under seismic and traffic hazards. *Earthquake Spectra*, 29(1), 127-153.

- Dong, Y., Frangopol, D. M., & Saydam, D. (2014). Pre-earthquake multi-objective probabilistic retrofit optimization of bridge networks based on sustainability. *Journal of Bridge Engineering*, 19(6), 04014018.
- Faturechi, R., & Miller-Hooks, E. (2015). Measuring the performance of transportation infrastructure systems in disasters: A comprehensive review. *Journal of infrastructure systems*, 21(1), 04014025.
- Galasso, C., et al. (2021). "Risk-based, Pro-poor Urban Design and Planning for Tomorrow's Cities." *International Journal of Disaster Risk Reduction*, 58, 102158.
- Gardoni, P., & Murphy, C. (2020). Society-based design: promoting societal well-being by designing sustainable and resilient infrastructure. *Sustainable and Resilient Infrastructure*, 5(1-2), 4-19.
- Hackl, J., Adey, B. T., & Lethanh, N. (2018). Determination of near-optimal restoration programs for transportation networks following natural hazard events using simulated annealing. *Computer-Aided Civil and Infrastructure Engineering*, 33(8), 618-637.
- He, X., & Cha, E. J. (2022). State of the research on disaster risk management of interdependent infrastructure systems for community resilience planning. *Sustainable and Resilient Infrastructure*, 7(5), 391-420.
- McDaniels, T. L., Chang, S. E., Hawkins, D., Chew, G., & Longstaff, H. (2015). Towards disaster-resilient cities: An approach for setting priorities in infrastructure mitigation efforts. *Environment systems and decisions*, 35, 252-263.
- Miller, M., and Baker, J. W. (2016). "Coupling mode-destination accessibility with seismic risk assessment to identify at-risk communities." *Reliability Engineering & System Safety*, 147, 60-71.
- Musakwa, W., and Gumbo, T. (2017). "Impact of urban policy on public transportation in Gauteng, South Africa: Smart or dumb city systems is the question." *Carbon Footprint and the Industrial Life Cycle: From Urban Planning to Recycling*, 339-356.
- Serulle, N. U., & Cirillo, C. (2014). Accessibility of low-income populations to safe zones during localized evacuations. *Transportation research record*, 2459(1), 72-80.
- Silva-Lopez, R., Bhattacharjee, G., Poulos, A., & Baker, J. W. (2022a). Commuter welfare-based probabilistic seismic risk assessment of regional road networks. *Reliability Engineering & System Safety*, 227, 108730.
- Silva-Lopez, R., Baker, J. W., & Poulos, A. (2022b). Deep learning-based retrofitting and seismic risk assessment of road networks. *Journal of Computing in Civil Engineering*, 36(2), 04021038.
- United Nations Office for Disaster Risk Reduction. (2015). *Sendai framework for disaster risk reduction 2015–2030*. UNDRR. <https://www.undrr.org/publication/sendai-framework-disaster-risk-reduction-2015-2030>
- Wein, A., Ratliff, J., Báez, A., & Sleeter, R. (2016). Regional analysis of social characteristics for evacuation resource planning: ARkStorm scenario. *Natural Hazards Review*, 17(4), A4014002.
- Yu, M., Chen, Z., Long, Y., and Mansury, Y. (2022). "Urbanization, land conversion, and arable land in Chinese cities: The ripple effects of high-speed rail." *Applied Geography*,

146, 102756.

Zhu, P. (2021). "Does high-speed rail stimulate urban land growth? Experience from China." *Transportation Research Part D: Transport and Environment*, 98, 102974.

Towards inclusive risk-informed infrastructure development in expanding cities

We thank the reviewer for taking the time to review the revised manuscript. Their comments are provided below in italic text, followed by our responses in normal text. Excerpts from the revised text are written in blue. Any line numbers mentioned refer to the clean version of the revised manuscript.

Reviewer #2

Q1.1 On balance, I am very comfortable with the changes made by the authors and the general form of the manuscript presently. I have one final comment prior to publication based on the review response documents:

Please integrate the extensive rebuttal you wrote for Q2.4 and Q2.5 in the review response document for reviewer 2 into the discussion of the manuscript. I am well convinced by the page-or-so response, but the few paragraphs which made it to the paper do not fully capture the nature of what you replied in the review rebuttal.

Reply: We appreciate the reviewer's positive assessment of our work.

We have further integrated our rebuttal for Q2.4 into the manuscript, as detailed in the following bullet points.

- The "Data description" section now provides further clarification that the willingness to pay calculation does not consider risk, as follows (lines 321 to 323): *"Note that these distances assume normal day-to-day infrastructure performance, such that a household's willingness to pay for a residential unit does not account for natural-hazard-induced travel disruptions"*
- The "Discussion" section now includes the following statement on the hypothetical scenario of wealthier people living in hazard-prone areas (lines 433 to 438): *"If more higher income households resided in flood-prone areas, the extent and pattern of gentrification would differ in line with the corresponding offset in the (generally larger) prices and values of willingness to pay; we do not investigate this hypothetical scenario in detail, because the strong spatial relationship between income and hazard exposure that exists within Tomorrowville has been deliberately designed to mirror real-world contexts.."*

We note that our rebuttal to this comment was already reflected in the following lines (424 to 433) of the "Discussion" section:

"Sensitivity analyses involving some of the case study parameters confirmed a complicated relationship between infrastructure performance and gentrification; better infrastructure performance does not necessarily entail an additional gentrification cost, underlining the importance of explicitly tracking both variables for informed decision making. Furthermore, these analyses revealed that low-income households would remain disproportionately susceptible to triggered relocations (and therefore would still benefit the most from a gentrification cap) even if rental prices increased somewhat (without considering any infrastructure expansion). It is important to note that these conclusions are based on one example involving the road infrastructure of a Global South virtual testbed in the presence of flooding, where there is a strong spatial correlation between income and level of hazard exposure."

We have now integrated our rebuttal for Q2.5 into the manuscript, such that the “Discussion” section now includes the following text on the assumptions associated with willingness to pay (lines 454 to 461): “.. *a household’s willingness to pay is exclusively determined by the travel distance to these critical infrastructure under normal conditions (i.e., disruptions from hazards are not considered to influence the perceived value of a given residence). While it is realistic to assume that residences with greater connectivity to important locations are valued higher^{1,2,3}, such that infrastructure development in hazard-prone areas could force local low-income households out of their homes, it may not be reasonable to presume that the attractiveness of better connected areas to higher income households is independent of exposure to natural hazards. However, tracking household movements beyond triggered relocations is outside the scope of this study.*”

References:

1. Shin, K., Washington, S., and Choi, K. (2007). “Effects of transportation accessibility on residential property values: Application of spatial hedonic price model in Seoul, South Korea, metropolitan area.” *Transportation Research Record: Journal of the Transportation Research Board*, 1994(1), 66–73
2. White, M. J. (1988). “Location choice and commuting behavior in cities with decentralized employment.” *Journal of Urban Economics*, 24, 129–152
3. Brandt, S., and Maennig, W. (2012). “The impact of rail access on condominium prices in Hamburg.” *Transportation*, 39(5), 997–1017